# Cellular population dynamics shape the route to human pluripotency

Francesco Panariello [1,14], Onelia Gagliano [2,3,4,14], Camilla Luni [5,6,14], Antonio Grimaldi[1,14], Silvia Angiolillo[2,3], Wei Qin[2,3,5], Anna Manfredi[1,13], Patrizia Annunziata[1,13], Shaked Slovin[1], Lorenzo Vaccaro[1], Sara Riccardo[1,13], Valentina Bouche[1], Manuela Dionisi[1], Marcello Salvi[1], Sebastian Martewicz [5], Manli Hu[5], Meihua Cui[5], Hannah Stuart[2,3], Cecilia Laterza [2,3], Giacomo Baruzzo [7], Geoffrey Schiebinger [8], Barbara Di Camillo [7,9,10], Davide Cacchiarelli [1,11,12,15] ✉ & Nicola Elvassore [2,3,4,5,15] ✉

Human cellular reprogramming to induced pluripotency is still an inefficient process, which has hindered studying the role of critical intermediate stages. Here we take advantage of high efficiency reprogramming in microfluidics and temporal multi-omics to identify and resolve distinct sub-populations and their interactions. We perform secretome analysis and single-cell transcriptomics to show functional extrinsic pathways of protein communication between reprogramming sub-populations and the re-shaping of a permissive extracellular environment. We pinpoint the HGF/MET/STAT3 axis as a potent enhancer of reprogramming, which acts via HGF accumulation within the confined system of microfluidics, and in conventional dishes needs to be supplied exogenously to enhance efficiency. Our data suggest that human cellular reprogramming is a transcription factor-driven process that it is deeply dependent on extracellular context and cell population determinants.

The discovery of human induced pluripotent stem cells (hiPSCs)[1] has emphasized the function of transcription factors in controlling cell identity, overlooking the role of cell-extrinsic signals. The reprogramming of somatic cells into hiPSCs is paradigmatic of a transcription factor-driven change of cell identity in three distinct and well-defined phases: cells exit a somatic state, transition through a transgene-dependent promiscuous transcriptional and epigenetic state, finally establish a self-renewing pluripotent identity. Several studies have established hallmarks and roadmaps of hiPSC formation[2–6], and new technological advancements, such as single-cell analyses, further refined our understanding of the reprogramming process[7,8]. These works better characterized initial and final stages of human reprogramming in detail but connected with intermediate stages via hypothetical and more uncertain trajectories. Whilst there is a body of literature describing reprogramming trajectory in mouse[9–11], the fine dynamics of human reprogramming intermediates, which

[1]Telethon Institute of Genetics and Medicine (TIGEM), Armenise/Harvard Laboratory of Integrative Genomics, Pozzuoli, Italy. [2]Department of Industrial Engineering, University of Padova, Padova, Italy. [3]Veneto Institute of Molecular Medicine (VIMM), Padova, Italy. [4]Stem Cell and Regenerative Medicine Section, GOS Institute of Child Health, University College London, London, UK. [5]Shanghai Institute for Advanced Immunochemical Studies (SIAIS), ShanghaiTech University, Shanghai, China. [6]Department of Civil, Chemical, Environmental and Materials Engineering (DICAM), University of Bologna, Bologna, Italy. [7]Department of Information Engineering, University of Padova, Padova, Italy. [8]Department of Mathematics, University of British Columbia, Vancouver, Canada. [9]Department of Comparative Biomedicine and Food Science, University of Padova, Padova, Italy. [10]CRIBI Biotechnology Center, University of Padova, Padova, Italy. [11]Department of Translational Medicine, University of Naples "Federico II", Naples, Italy. [12]School for Advanced Studies, Genomics and Experimental Medicine Program, University of Naples "Federico II", Naples, Italy. [13]Present address: NEGEDIA (Next Generation Diagnostic srl), Pozzuoli, Italy. [14]These authors contributed equally: Francesco Panariello, Onelia Gagliano, Camilla Luni & Antonio Grimaldi. [15]These authors jointly supervised this work: Davide Cacchiarelli, Nicola Elvassore. ✉e-mail: d.cacchiarelli@tigem.it; nicola.elvassore@unipd.it

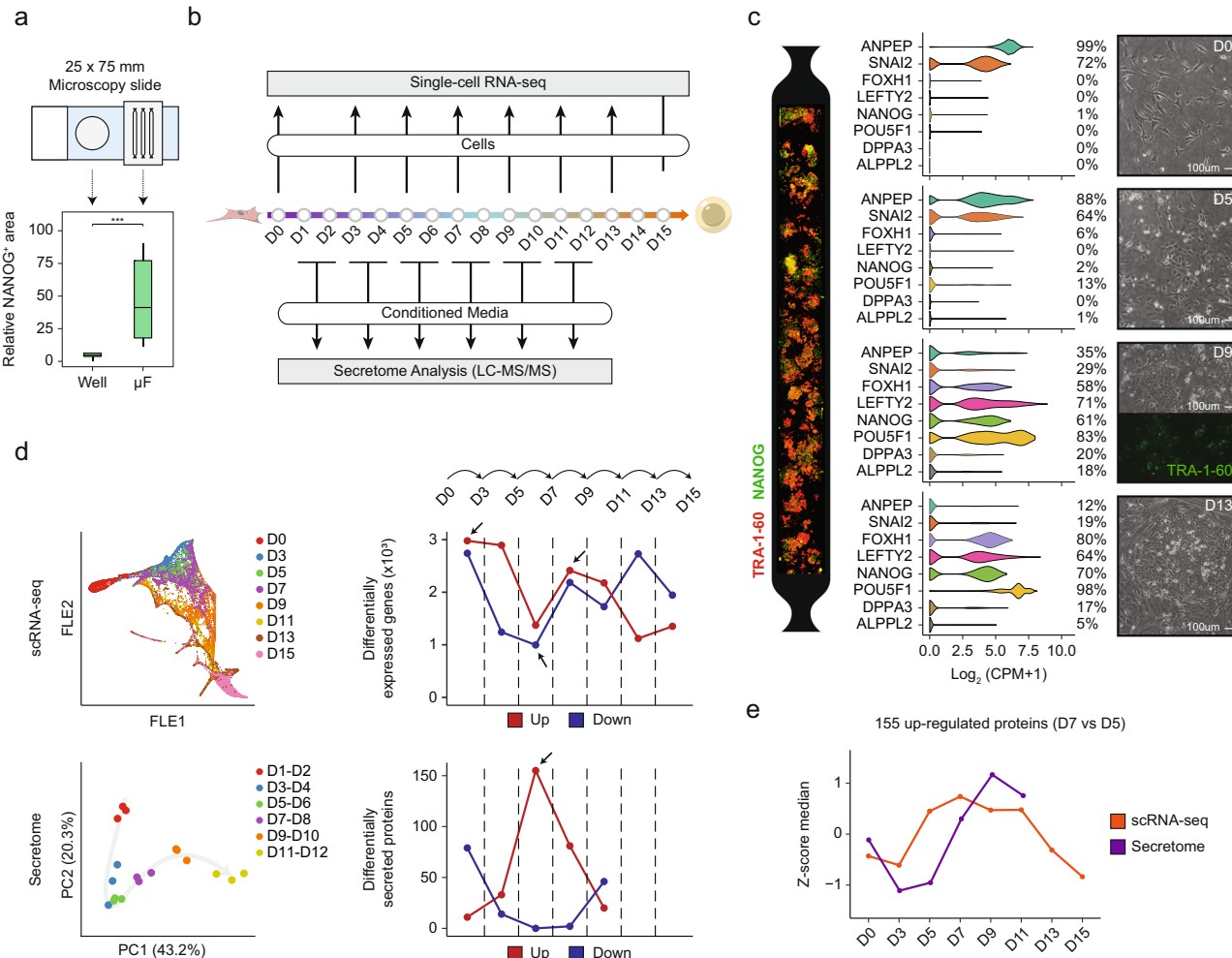

**Fig. 1 | Development of a temporal multi-omic approach to study human cell reprogramming in microfluidics. a** Schematics of the in-scale conventional (Well) and microfluidic (μF) setup (top), and comparison of reprogramming efficiency therein (bottom). Two-sided Wilcoxon's test was used to assess differences among the conditions. $n = 8$ for Well and $n = 15$ for μF (***$P = 0.0001593$). **b** Experimental design for -omics experimental data collection during reprogramming in microfluidics (Methods). scRNA-seq data were collected by stopping parallel experiments at day 0, 3 and every 48 h. Proteomic data were obtained by tandem mass spectrometry analysis of conditioned media along the same reprogramming experiments. For accurate relative quantification between time points peptides were labeled by TMT. **c** Left: immunostaining of a single microfluidic channel at day 14 for pluripotency markers (NANOG and TRA-1-60). Middle and right: Morphological (right) and transcriptional (middle) changes occurring during reprogramming,

sampled at day 0 (D0), day 5 (D5), day 9 (D9) and day 13 (D13). The percentage of cells expressing a particular gene marker in a certain time-point is reported. CPM: counts per million. **d** Left: dimensionality reduction plots for scRNA-seq (FLE) and Proteomic (PCA) data. Sequencing data is shown as the distribution of transcriptional patterns for single cells across sampled time-points. Proteomic data is shown as the distribution of the proteomic pattern for sampled conditioned medium over a 48-hour period. Right: absolute number of differential features for each -omics data, both up- (Up - red) and down-regulated (Down - blue). Each value refers to the differential analysis between subsequent time-points. Peaks of deregulation are highlighted (arrows). **e** Median z-score of the 155 proteins found up-regulated from day 5 (D5) to day 7 (D7) in proteomic data. Trends have been evaluated along the time-course for both -omics data.

constitute the bottleneck of the process, remain largely unexplored due to the complexity of recognizing and selecting rare phenotypes that will evolve into a hiPSC fate.

It is thought that individual cells during reprogramming evolve with a smooth progression under a selective pressure that results into dominant "elite" clones[12–15]. It has been recently suggested that reprogramming of murine cells may also depend on population dynamics through cell-non-autonomous mechanisms in a context-dependent manner, i.e. mediated by cell-secreted factors[10,16,17]. Consistently with this hypothesis, we recently reported that the efficiency of reprogramming of human somatic cells to hiPSCs can be dramatically improved in a microfluidic confined environment[18–20], which enhances the accumulation of secreted factors[21–26] and sustains the acquisition of both primed[18,19] and naive human pluripotency[27].

In line with this, we hypothesize that during human cellular reprogramming, specific subpopulations control fate decisions towards pluripotency by cell-extrinsic factors. We envision that the communication between distinct intermediate sub-populations and their shared extracellular environment lying in-between contributes to shaping the route to pluripotency.

In this work, we take advantage of reprogramming in microfluidics to have a high efficiency within a confined environment (Fig. 1a), where secreted signals are accumulated, and distinctive intermediate sub-populations can be effectively captured and characterized. We perform integrated temporal multi-omic profiling during reprogramming to reveal finely regulated dynamics of secreted proteins accumulating in the extracellular space and a cellular heterogeneity arising during intermediate stages of reprogramming. We investigate how these complex population dynamics are modulated by

extrinsic signals and how this facilitates the prompt acquisition of pluripotency.

## Results

### Development of a temporal multi-omic approach to study human cell reprogramming in microfluidics

In order to dissect cellular heterogeneity arising during human somatic cell reprogramming and the role of the surrounding micro-environment, we combined high-efficiency reprogramming (Fig. 1a) with high-throughput single-cell RNA (scRNA-seq) and tandem mass spectrometry (LC-MS/MS) on conditioned media (Fig. 1b). High-efficiency reprogramming of human fibroblasts was achieved in microfluidics (uF) with daily transfections of non-modified messenger RNAs (mRNAs) encoding for POU5F1 (OCT4), SOX2, KLF4, MYC, LIN28, NANOG [18,19] (Methods). With respect to non-uF methods, this approach generates a more efficient reprogramming, with a considerably and significantly higher number of pluripotent colonies retrieved at the end of the process (Fig. 1a). The analysis of the secretome was performed on conditioned media pooled from microfluidic channels every 2 days (Fig. 1b bottom)[18,21,22]. To maximise the effectiveness of identifying the endogenous secreted proteins we used a chemically defined medium based on E6 medium with the addition of FGF2[19] which has been shown to preserve the high efficiency of microfluidic reprogramming (Supplementary Fig. 1a) while enabling high-resolution and accurate detection of cell-secreted proteins (Supplementary Fig. 1B–D and Methods). We quantified 4542 proteins, the majority identified in 3 replicates (81%) and the others identified in only 2 replicates. Protein labeling by tandem mass tags (TMTs)[28] allowed us to obtain a relative quantification of each protein along the process (Supplementary Fig. 1A). On the other hand, cells recovered for scRNA-seq were collected before the first transfection (D0), 3 days after transfection (D3) and then every 2 days (D5-D15—Fig. 1b top). We generated sequencing libraries from independent captures for at least two replicates per time-point, collecting altogether more than 40,000 single-cell transcriptomes. After the evaluation of the sequencing quality, cell filtering, down-sampling and feature filtering, the final datasets consisted of 20,000 high-quality single-cell transcriptome for a total of 12,932 features detected (Supplementary Fig. 1E, F and Methods). We have already reported that morphological changes associated with conventional human cell reprogramming are recapitulated in our microfluidic systems[19]: we observe quick mesenchymal to epithelial transition (MET - before day 5) and epithelial cells clustering and hiPSCs colony formation as soon as day 9 (Fig. 1c left and right). Additionally, we correlated gene expression variation of known markers to these changes, observing a bimodal, progressive loss of fibroblast markers expression (ANPEP, SNAI2) and the opposite trend for developmental patterning genes (FOXH1 and LEFTY2), that we[2] and others[3] previously identified. (Fig. 1c middle). The expression of these genes anticipated the onset of canonical pluripotency-related genes (POU5F1 and NANOG - Fig. 1c middle). Interestingly, although expressed by fewer cells (20% at D9), also pre-implantation genes (DPPA3 and ALPPL2) showed a transient activation during intermediate stages, decreasing when epiblast pluripotency (lower NANOG and higher POU5F1) is reached, as we have already described[2] (Fig. 1c middle and Supplementary Fig. 1H).

To test the hypothesis that both cellular and extracellular dynamics are interconnected, we reduced data dimensionality. scRNA-seq data dimensionality was reduced using non-linear algorithms, namely force layout embedding (FLE). The resulting diagram (Fig. 1d top left) illustrates the profile of each cell as a point in a Euclidean space where cells are grouped based on their transcriptional similarity. Interestingly, the graph shows high homogeneity of the fibroblasts population at day 0 (D0) and higher heterogeneity thereafter, with cells placed in the space according to their sampling day. Secretome data dimensionality was reduced via a principal component analysis

that showed samples following a reprogramming temporal trajectory, in agreement with sequencing data, with high reproducibility between replicates (Fig. 1d bottom left). We finally compared the differential features of each dataset along the time (Fig. 1d top right). As expected, during the transition from D0 to D3, gene expression was the most influenced by the transfection of reprogramming factors, as evidenced by the high number of differentially expressed genes. From D3 on, transcriptional changes start to decrease until D7, where they reach the minimum magnitude. Finally, we observed at D7-D9 and D11-D13, two more transcriptional waves in line with the onset of developmental transitions and final acquisition of pluripotency.

Notably, in between the two first transcriptional waves (from D5 to D7), we observed a great increase in the number of secreted proteins (Fig. 1d bottom right). We reason that the initial massive changes in gene expression might induce the specification of a set of secreted molecules that becomes manifest in the medium at D7 (Fig. 1e).

The massive number of secreted proteins at D5-D7 pointed us to investigate the quality of secreted proteins and cell population dynamics occurring in such a peculiar window of time.

### Embryonic ECM accumulates during reprogramming

We specifically selected 555 proteins known to be secreted (Supplementary Data 1) to get rid of intracellular proteins potentially released by dead cells. We classified the identified categories into the two broad groups of extracellular matrix (ECM)- and soluble signal-related functional annotations (Fig. 2a and Supplementary Dataset 2). Many ECM-related categories were highly significant, including ECM deposition, degradation and remodeling, and both integrin- and non-integrin-mediated cell-ECM interactions (Fig. 2a left). A previous RNAi screen also identified the critical role of cell adhesion in human reprogramming, highlighting the role of intercellular factors needed for filament assembly, branching, and disassembly[5].

In our data, we found an overall increasing trend of ECM-related protein accumulation, with different ECM components exhibiting distinct dynamics (Fig. 2b). These dynamic changes started already at days 3-4 (SPP1, COL4A1/2, SPARC), in some cases at days 5-6 (LAMC1), or even later (COL18A1). We wondered whether the observed global changes somehow resembled embryo development stages. To address this question, we selected the ECM proteins in our data that were previously reported to be expressed at mRNA level at different stages of human embryo development[29]. The concentration dynamics of these proteins in our system showed the progressive establishment of an ECM that recapitulates the one deposited at the stage of the late inner cell mass (Fig. 2b, Supplementary Data 3). In conclusion, our data support the idea that during reprogramming, not only fibroblasts are converted to a primed pluripotent phenotype, but also the extracellular context is shaped accordingly.

### Dynamics of extrinsic regulatory signals during reprogramming

Our secreted proteins were enriched in several other processes, demonstrating that this extracellular environment is rich in regulatory signals. Figure 2a (right) shows a selection of signalling pathways enriched within the Reactome database (see Supplementary Data 2 for full results). Among receptor tyrosine kinase pathways, PDGF and WNT have already been shown to be implicated in embryo development and reprogramming[30,31]. We also identified the MET pathway as a link between cell-cell communication via soluble environment, and cell-ECM interaction via PTK2 (also known as FAK) adhesion. Moreover, the regulation of insulin-like growth factor (IGF) pathway through IGF binding proteins (IGFBPs) was significantly enriched, in line with previous studies[32].

Looking at the temporal profiles of enriched signalling pathway proteins and ligands (Fig. 2c, Supplementary Data 4), we found a progressive accumulation of proteins that were previously shown to play a role in mouse cell-non-autonomous reprogramming regulation:

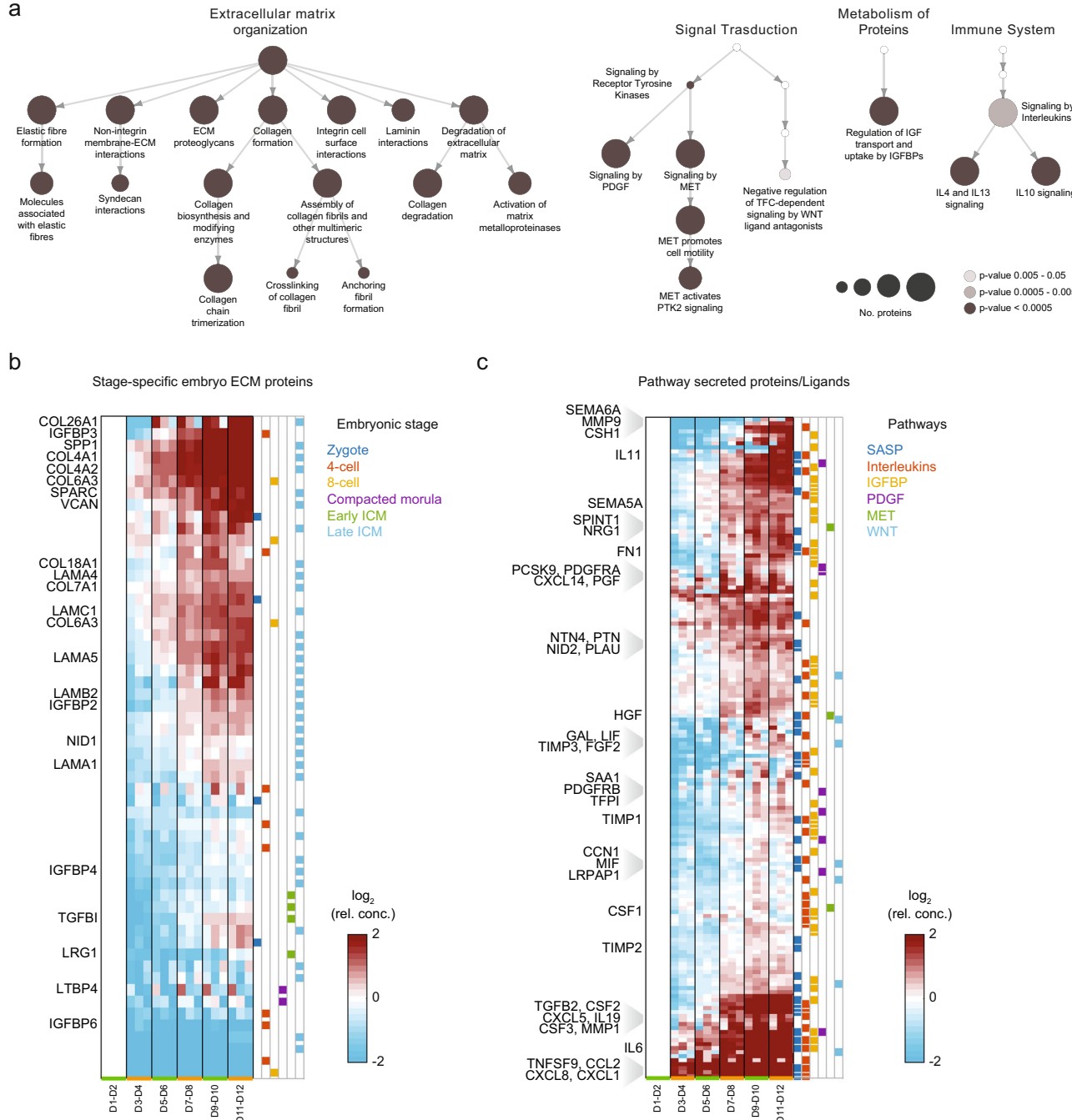

**Fig. 2 | Proteomic analysis of cell-secreted proteins demonstrates a rich extracellular signalling environment along human fibroblast reprogramming.** **a** Enrichment analysis within the Reactome database of the 555 proteins identified as secreted (Supplementary Data 1). Edges connecting different categories reproduce Reactome hierarchy relationships. Complete results are reported in Supplementary Data 2. **b** Hierarchical clustering of proteins identified in this study and belonging to the core ECM components[36] at specific stages of embryo development[29]. **c** Hierarchical clustering of secreted proteins from the following enriched signalling pathways (according to Reactome database): Signaling by Interleukins (R-HSA-449147), Regulation of Insulin-like Growth Factor transport and uptake by Insulin-like Growth Factor Binding Proteins (R-HSA-381426), Signaling by PDGF (R-HSA-186797), Signaling by MET (R-HSA-6806834), Signaling by WNT (R-HSA-195721); playing a role as senescence-associated secreted proteins[60–63], and as ligands[39]. Collagens and Laminins were excluded from these protein sets as they were plotted in **b**. Only selected protein names discussed within the text or identified as most specific cluster markers in the subsequent single-cell RNA-seq analysis are shown.

some senescence-associated secreted proteins (SASP), such as CXCL1 (also known as Gro-α), CXCL8, CCL2, IL6[33]; YAP-target CCN1, also known as CYR61[34]; inflammatory cytokines, such as IL6/11/19, CSF1/2/3, LIF[17]. We found that JAK-STAT pathway, downstream of interleukin signalling, was also significantly differentially expressed at transcriptomic level between freshly-derived microfluidic hiPSC colonies

and the same colonies after 3-passage expansion in conventional wells[18] (Supplementary Fig. 2A).

We conclude that secreted proteins follow precise dynamics during reprogramming and encompass a number of potential regulators of autocrine/paracrine signalling, including those involved in ECM-mediated and soluble communication.

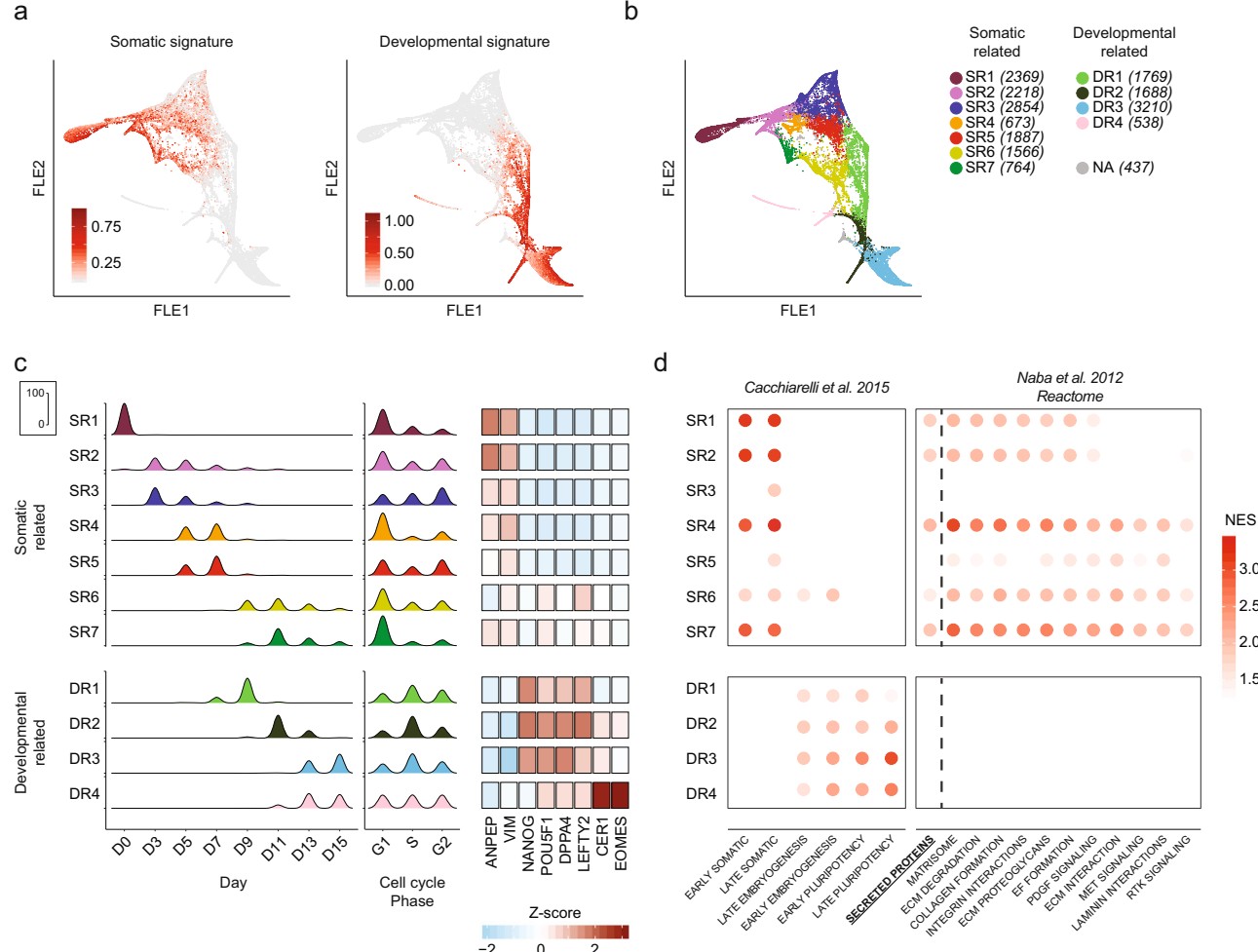

**Fig. 3 | Single-cell RNA-seq analysis of human reprogramming cells unveils a dynamic somatic subpopulation involved in the expression of signalling and ECM related genes. a** Somatic and Developmental signatures enrichment scores shown along the FLE map. **b** FLE map showing the distribution of cells across identified clusters. **c** Time-points and cell-cycle phase distribution for each cluster (left) and heatmap of Z-scored normalized counts, averaged by clusters, for key reprogramming related genes (right). NA cluster not shown. **d** GSEA results for each cluster. Only significant results are shown. The gene set made of the secreted proteins found in this work is written in bold. NES, Normalized enrichment score.

## Resolving cell population heterogeneity during reprogramming

As single-cell -omics succeeded in detecting heterogeneity arising during human cell reprogramming, we applied an unsupervised community detection algorithm[35] to our scRNA-seq data identifying 12 cell clusters. We then took advantage of our formerly defined reprogramming-associated gene signatures[2] to annotate them (Fig. 3a, b). 7 clusters showed high expression of somatic genes ("Somatic-Related" clusters, SR), whereas 4 clusters were highly enriched by the developmental signature ("Developmental-Related" clusters, DR). Finally, a residual cluster was not enriched by either of those signatures and it was characterized by a lower number of detected genes and total UMI counts, thus we named it "NA" and excluded it from further analyses (Fig. 3c, d and Supplementary Fig. 3A, B). As expected, SR clusters included non-transfected fibroblasts (SR1) and cells captured at earlier days (SR2-5), while DR clusters were enriched by cells collected at later time points (from D9 to D15) and highly cycling (Fig. 3c). However, more than 97% of SR6 and SR7 cells were sampled from day 11 (Fig. 3c) and were characterized by low but detectable expression of embryonic genes (e.g. *POU5F1*, *LEFTY2*) and were negative for *NANOG*, indicating reshaping of fibroblast identity but at the same time inefficient acquisition of pluripotency. Furthermore, these cells are in the G0/G1 phase of the cell cycle, thus confirming their somatic nature and suggesting peculiar identity in the reprogramming timeline (Fig. 3c). Despite their

developmental features, DR4 cells also did not express *NANOG*, while showing high and very specific transcriptional levels of mesendoderm genes (e.g. *CER1*, *EOMES*), suggesting a possible similarity with a differentiating stage. DR clusters display higher expression of embryonic-related signatures[29], thus they appear to contain the productively reprogramming cells (Supplementary Fig. 3C). However, the role of the SR clusters is less clear (Fig. 3d left). To address the role of SR clusters we perform Gene Set Enrichment Analysis (GSEA) using the secreted proteins previously identified (Fig. 3d right in bold) and some gene signatures that were found enriched in the proteomic analysis (Fig. 3d right and Supplementary Data 5). Surprisingly, the secreted proteins detected by mass spectrometry appear to be transcribed by the cells in the SR clusters, except for SR3 that might not be involved in the secretory phenotype. These results highlight the presence of an unproductive somatic fate, whose role is to express and secrete those factors that we found to be shaping the extracellular environment during reprogramming and that have been found to characterize later stages of embryonic development.

## Signalling contributions from different cellular subpopulations

Among all the gene sets analyzed, Matrisome[36] and Late pluripotency[2] associated genes were found to best describe the phenotype of D13-15 endpoints (Fig. 4a). Therefore, we decided to computationally

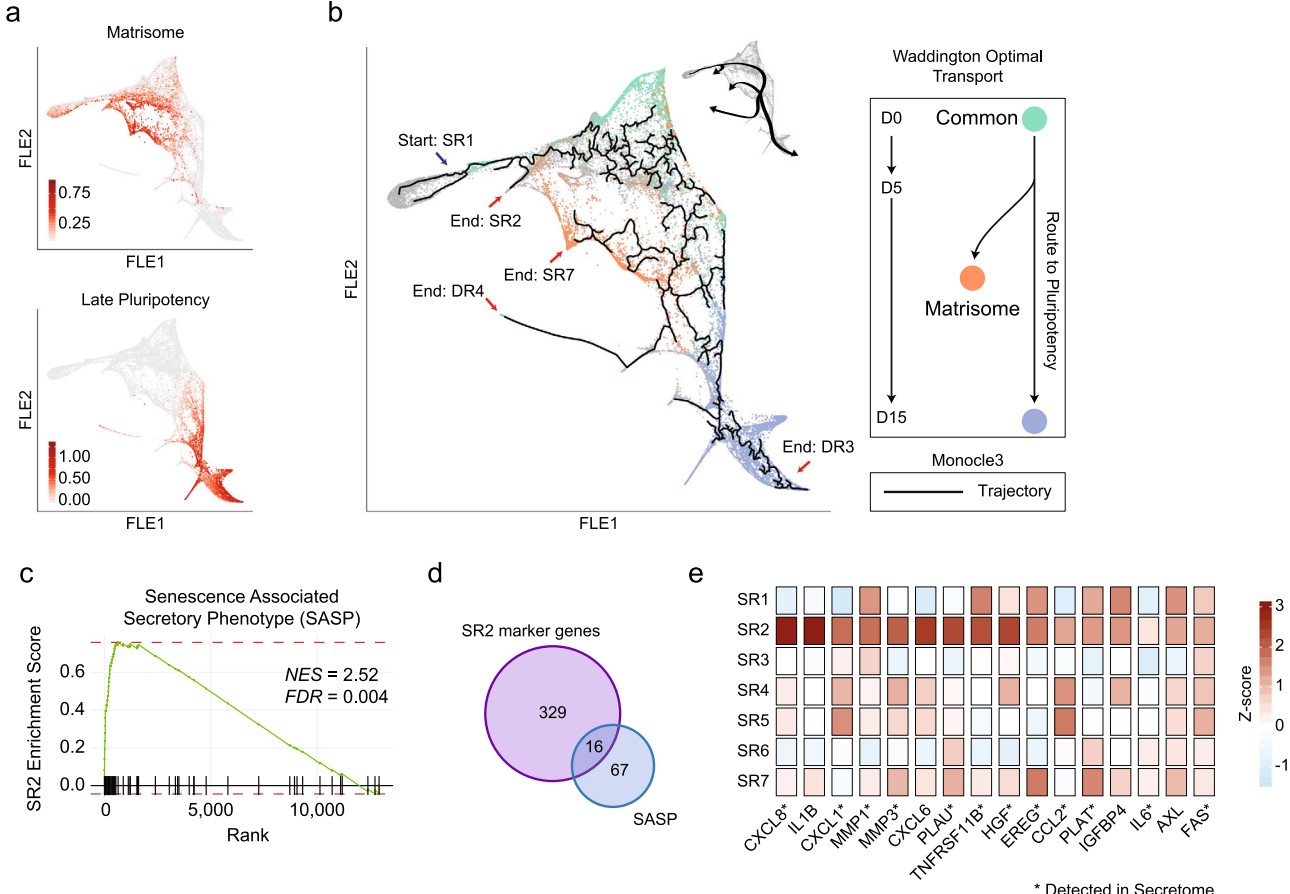

**Fig. 4 | Trajectory inference reveals different fates during reprogramming.**
**a** Matrisome and Late pluripotency enrichment scores shown along the FLE map.
**b** Monocle3 (black line) and WOT (colored dots) trajectory inferences are displayed on the FLE graph. Arrows point to the starting point (blue) and 4 end points (red) of the inferred trajectories. A representative scheme of the trajectories is shown on the top-right. **c** Enrichment Score graph relative to the GSEA of SR2 cluster for senescence-associated secreted proteins geneset (SASP)[60–63]. Black lines on the x axis represent a match between the ranked list and the geneset analyzed. NES, Normalized enrichment score. FDR, False Discovery Rate. **d** Venn diagram representing the intersection between SASP geneset and SR2 cluster marker genes and their relative gene expression, shown in a (**e**) heatmap of Z-scored normalized counts, averaged by clusters. Genes with (*) have been detected in secretome analysis.

investigate the routes linking such states to the somatic start-point by applying Waddington Optimal Transport (WOT)[10] (Fig. 4b and Supplementary Fig. 4A). Results showed a common path until day 5 (D5), after which cells started to exhibit different trajectories (Fig. 4b, Supplementary Fig. 4B). We validated these findings through an unsupervised pseudotime-based approach using Monocle3[37,38], which not only confirmed the bifurcation at day 7 (D7) leading to endpoints inside SR7 matrisomal and DR3 pluripotent clusters, but also introduced two additional outcomes inside DR4 and SR2, respectively (Fig. 4b and Supplementary 4C). While the mesendodermal nature of DR4 was previously assessed, we focused on the characterization of SR2. GSEA using common pathways (Methods) revealed the enrichment for terms related to signalling molecules (Supplementary Data 6), therefore, we hypothesized that this cluster might be implicated in the secretion of the ligands detected in the medium. Indeed, most of them were significantly enriched, with SASP having the highest enrichment score (Fig. 4c and Supplementary Fig. 4D). We found SASP genes are highly expressed and specific of this cluster (Supplementary Data 7), such as cytokines (*CXCL1*, *IL1B*, *CXCL8*), metalloproteases (*MMP1*, *MMP3*), *HGF* and its activators, *PLAU* and *PLAUR* (Fig. 4d, e). Notably, almost all of them were detected by LC-MS/MS with some (CXCL1, CXCL8, CCL2, SPP1, PLAU) being the first to be accumulated in the medium (Fig. 2c).

In conclusion, we were able to define human somatic reprogramming as a process consisting of two major outcomes, matrisomal

and pluripotent, deriving from the same starting cells which bifurcate around day 7 (D7). Moreover, among matrisomal somatic cells, we identified and characterized an early sub-population of cells which contributes to the expression and secretion of SASP-related signalling molecules.

## Reprogramming fates interact through different ligand-receptor pairs

To rationally understand whether somatic subpopulations arising during reprogramming are actively involved in the population crosstalk with productive reprogramming intermediates, we developed a ligand-receptor interaction analysis from the cells laying on the somatic trajectory towards the reprogramming ones (Fig. 5a). Using the previously identified secreted proteins (Fig. 1) that fall in the list of experimentally validated ligand-receptor couples[39], we restricted the number of putative interactors involved in subpopulation crosstalk to a set of 82 pairs (Supplementary Fig. 5A, Supplementary Data 8 and Methods). We were able to identify a standardized interaction score (sIS) by leveraging the gene expression trends of ligands along the matrisome route and of receptors along the path to pluripotency (Methods).

The results showed that almost every ligand-receptor pair had a significant sIS in at least one time-point (Supplementary Fig. 5B and Supplementary Data 9). Moreover, when looking at the couples with

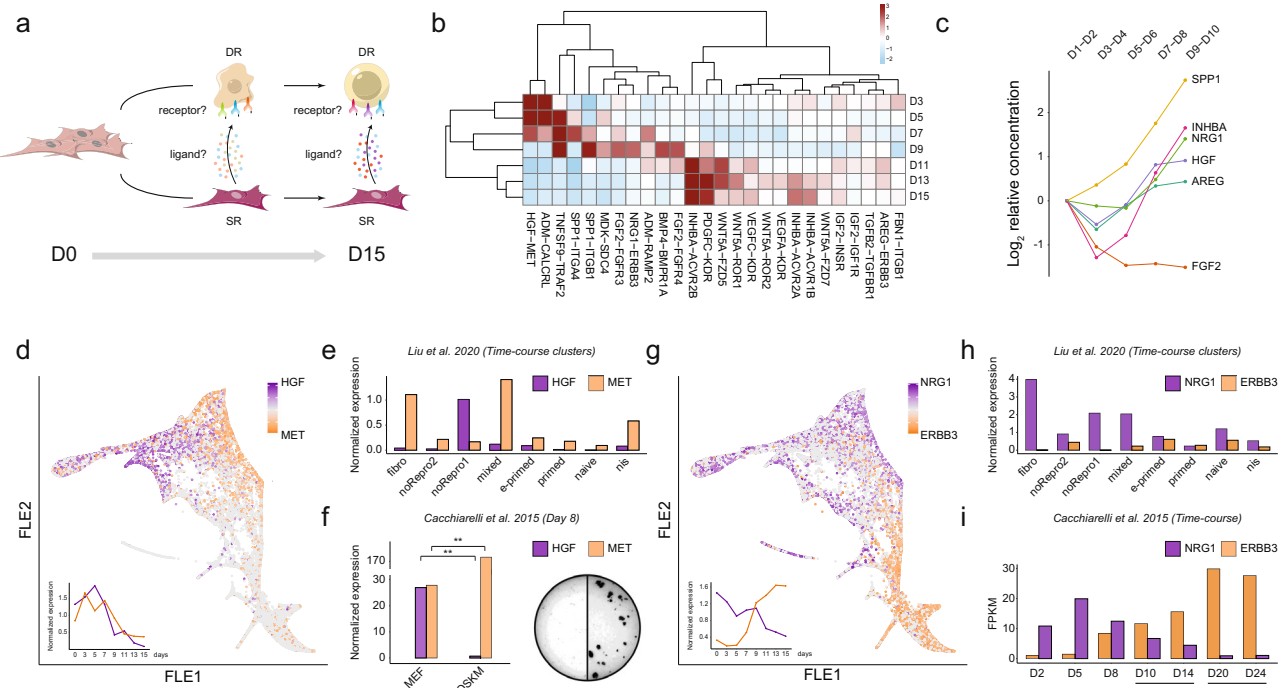

**Fig. 5 | Gene expression-based interaction analysis suggests an existing crosstalk between somatic and reprogramming cells through known and unknown ligand-receptor couples. a** Schematic representation of ligand-receptor interactions hypothesized during reprogramming. Fibroblasts (D0, left) develop two fates: a somatic secretory phenotype (bottom) and induced pluripotency (top). Black arrows show the directionality of the examined interaction. **b** Heatmap of z-scored standardized interaction scores for top ligand-receptor pairs. Selection criteria is described in Methods. **c** Log2 proteomic expression relative to D1-D2 of SASP-related ligands among the top pairs at each time-point. **d–f** HGF and MET gene expression profiles (log2 CPM) are shown in different reprogramming systems. **d** In our data, they are displayed on the FLE map as fold change relative to HGF and averaged across the time course (bottom-left). **e** In Liu et al., 2020[7], they are shown as averaged across their identified clusters. **f** Left: in Cacchiarelli et al., 2015[2], they are shown as mouse and human mean normalized expression at sampling day 8 (** BH-adjusted $p$-value <0.01). Right: Representative pictures of HiF-T DOX secondary reprogramming performed with or without depletion of MEFs in standard 12-well plates, assessed by immunostaining of TRA-1-60. **g–i** NRG1 and ERBB3 gene expression profiles are shown in different reprogramming systems. **g** In our data, they are displayed on the FLE map as fold change relative to NRG1 and averaged across the time course (bottom-left). **h** In Liu et al., 2020[7], they are shown as averaged across their identified clusters and (**i**) in Cacchiarelli et al., 2015[2], they are shown as mean FPKM across the time-course.

the greatest scores, we observed many ligands involved in signalling cascades which are already known to be associated with pluripotency maintenance, such as *Wnt, Tgfβ* and *Inhb* signalling[31,40,41] (Fig. 5b). These results were overall confirmed by a complementary unbiased approach, based on an alternative interaction score computed as a function of the absolute ligand and receptor expression levels and their $\log_2$ fold change with respect to the average expression level across all time points (Supplementary Fig. 5C, Supplementary Data 10 and Methods).

Among these interactors, 8 ligands were related to SASP, of these 4 were soluble and highly dynamic in both transcriptomic and proteomic data: SPP1, INHBA, NRG1 and HGF (Fig. 5c). As INHBA is a known pluripotency regulator[40], and SPP1 is the major HGF-regulated gene[42], we focused our analyses on HGF and NRG1.

The HGF-MET interaction occurred at early time-points of the reprogramming (Fig. 5d) with HGF expressed by cluster SR2 and SR5 and its receptor MET expressed by cluster DR1. Both HGF and MET were highly expressed in the early intermediate stages and decreased in the later time points, suggesting a role in the reprogramming intermediates. Thus, we explored whether the same HGF-MET dynamics was present in a conventional (i.e., Petri dish) human reprogramming approach[7] and not strictly related to the microfluidic environment. scRNA-seq data exploration, using authors-defined clusters[7], showed that the cluster noRepro1, enriched for SR signatures (Supplementary Fig. 5D), expressed high levels of HGF (Fig. 5e), whereas MET expression was observed in the mixed intermediate cluster, overlooked by the authors (Fig. 5e). Remarkably, the

analysis of RNAseq data from reprogramming of secondary human fibroblasts cultured on mouse embryonic fibroblast feeder (MEF)[2], showed the expression of HGF only from MEFs while MET was upregulated in human cells undergoing reprogramming at day 8 (OSKM -Fig. 5f left). Therefore, we performed reprogramming experiments with depletion or addition of MEFs and observed a drastic reduction in the ability of generating pluripotent colonies when cultured in absence of feeder cells (Fig. 5f right), suggesting a pivotal role of HGF-MET interaction in sustaining pluripotency. These results showed a common behaviour of HGF vs MET expression in the early phase of the reprogramming, being expressed by matrisome producing/supporting cells and reprogramming intermediates respectively, regardless of reprogramming approach and culture system. On the other hand, the NRG1-ERBB3 interaction showed higher sIS between clusters along the same developmental trajectory in a sequential fashion: NRG1 is expressed by DR clusters at earlier stages (until D9), while its receptor, Erb-B2 Receptor Tyrosine Kinase 3 (ERBB3), is expressed by late DR clusters (starting from D7) (Fig. 5g). The same information can be retrieved from Liu et al., 2020[7] and Cacchiarelli et al., 2015[2], observing the sequential expression of NRG1 then ERBB3 only along the reprogramming intermediates, with NRG1 decreasing halfway during reprogramming route, ERBB3 increasing from halfway, and a central timeframe of co-presence (Fig. 3h, i). Therefore, as NRG1-ERBB3 expression occurs only along the reprogramming trajectory, we did not get significant results when comparing MEFs versus human reprogramming intermediates from our human secondary system (Supplementary Fig. 5E)[2].

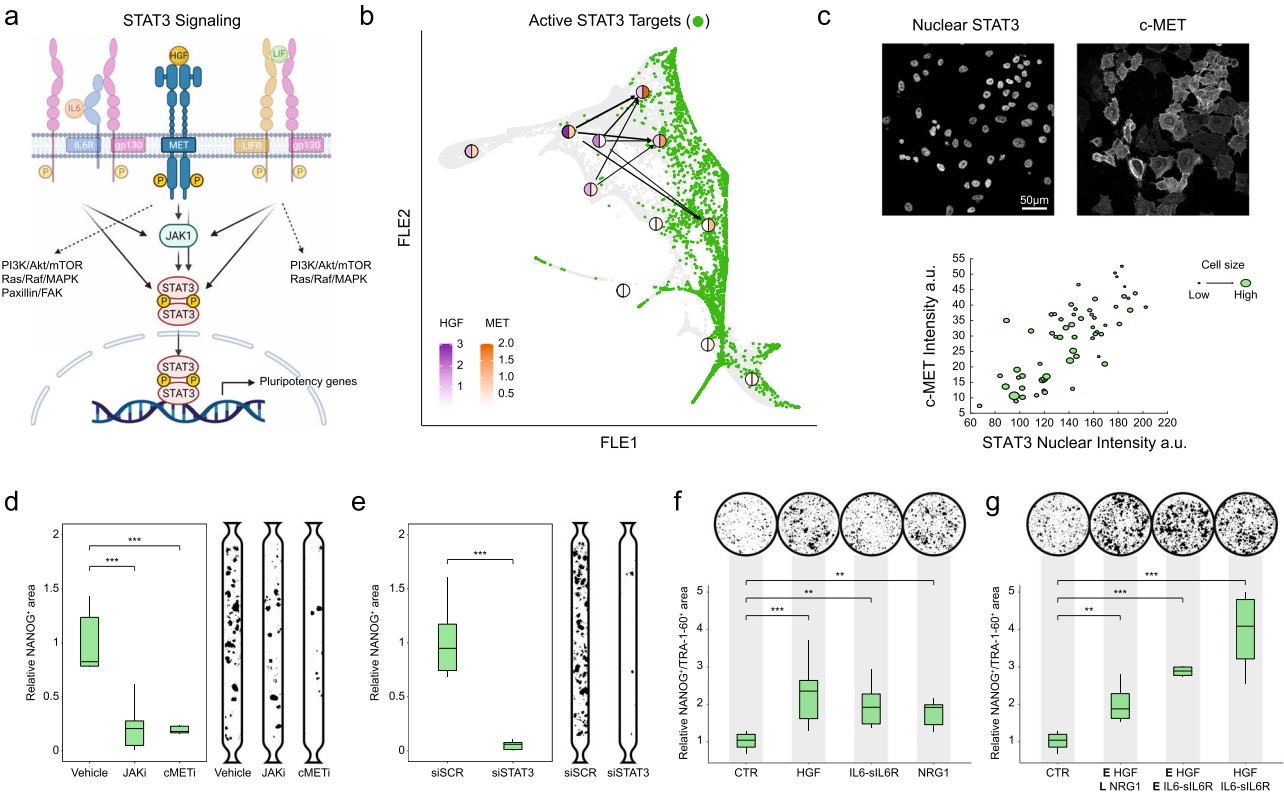

**Fig. 6 | Perturbation of STAT3 pathway components affect the efficiency of reprogramming. a** A schematic representation of HGF/c-MET/STAT3 signalling pathway (Created with BioRender.com). **b** STAT3 target expression correlates with MET transcription. In the FLE graph, green dots represent cells with positive enrichment scores for STAT3 target genes (Methods). Bigger circles summarize averaged HGF (left) and MET (right) gene expression in identified clusters. Significant inter-cluster HGF-MET interactions are displayed (arrows). Arrow thickness relates to the strength of the interaction. **c** Top, representative images of expression of nuclear STAT3 and c-MET during reprogramming performed in microfluidics at day 6. Bottom, correlation between the expression intensity of nuclear STAT3, c-MET, and cell size obtained from experimental data shown on top. Data from $n = 61$ cells ($n = 3$ independent experiments). **d** Left, reprogramming efficiency in microfluidics measured as the relative area occupied by NANOG+ colonies in cells upon inhibition of c-Met and JAK1 kinases using small molecules at day 12, compared to the ones treated with the vehicle ($n = 6$ for vehicle, $n = 12$ for JAKi and $n = 7$ for c-METi); ANOVA followed by two-sided Dunnett's multiple comparisons test was used to assess differences among the conditions (JAKi − 95% CI [0.5645,1.039], *** FDR = 0.0001; cMETi − 95% CI [0.548, 1.076], *** FDR = 0.0001). Right, representative quantification pictures in microfluidic channels assessed by immunostaining of NANOG. **e** Left, reprogramming efficiency in microfluidics upon knockdown of STAT3 using siRNAs at day 12 ($n = 8$ for scramble siRNA, $n = 11$ for siSTAT3);

two-sided unpaired *t*-test was used to assess differences among the conditions (95% CI [0.7511,1.154], ***$P < 0.0001$). Right, representative quantification pictures in microfluidic channels assessed by immunostaining of NANOG. **f** Bottom, reprogramming efficiency in standard 24-well plates upon addition of HGF, IL-6 and soluble IL6 receptor (sIL6R), or NRG1 at day 9 ($n = 14$ for control, $n = 19$ for HGF, $n = 5$ for IL6 + sIL6R, $n = 16$ for NRG1); ANOVA followed by two-sided Dunnett's multiple comparisons test was used to assess differences among the conditions (HGF − 95% CI [−1.693, −0.6466], *** FDR = 0.0001; IL6 + sILR − 95% CI [−1.744, −0.1955], **FDR = 0.0083; NRG1 − 95% CI [−1.319, −0.2312], ** FDR = 0.0021). Top, representative quantification pictures in standard 24-well plates assessed by immunostaining of NANOG and TRA-1-60. **g** Bottom, reprogramming efficiency in standard 24-well plates upon temporally modulate addition of HGF, IL6 and soluble IL6 receptor (sIL6R), and NRG1 at day 9 ($n = 14$ for control, $n = 6$ for HGF in the early phase and NRG1 in the late phase, $n = 4$ for HGF + IL6 + sIL6R in the early phase and NRG1 in the late phase, $n = 4$ for HGF + IL6 and sIL6R + NRG1 for the entire process); ANOVA followed by two-sided Dunnett's multiple comparisons test was used to assess differences among the conditions (E HGF + L NRG1 − 95% CI [−1.705, −0.2549], ** FDR = 0.0038; E HGF/IL6 + sILR + L NRG1 − 95% CI [−2.698, −1.013], *** FDR = 0.0001; ALL − 95% CI [−3.746, −2.061], *** FDR = 0.0001). Top, representative quantification pictures in standard 24-well plates assessed by immunostaining of NANOG and TRA-1-60.

---

Altogether, these findings suggest a crosstalk between cell subpopulations, with an active role of non-pluripotent cells in supporting the route of other cells to pluripotency. We demonstrated that such non-pluripotent cells can be part of the same (i.e. NRG1 and ERBB3 both expressed during DR trajectory to pluripotency) or different trajectories (i.e. HGF ligand expressed by SR trajectory towards matrisome vs MET receptor expressed by DR trajectory towards pluripotency).

### HGF-MET crosstalk functionally sustains the acquisition of pluripotency through STAT3

Considering the results from the ligand-receptor analyses, we then asked whether the HGF-MET interaction has a functional role in the progression of intermediate states towards pluripotency.

HGF is a growth factor involved in many cell functions and it is mostly secreted by mesenchymal cells, while acting on epithelial ones[43]. In our reprogramming, it is biologically active as its activator complex PLAU/PLAUR was also found in the secreted medium (Fig. 2c).

On the other hand, MET is a tyrosine kinase receptor activated by its ligand HGF. This binding induces MET catalytic activity and results in downstream initiation of multiple pathways, including STAT3 direct phosphorylation or via Janus kinase 1 (JAK1 - Fig. 6a). This activation axis is shared with other two ligands (i.e. LIF and IL6), known to be involved in murine pluripotency (Fig. 6a)[16,44].

To test the STAT3 pathway involvement in our reprogramming setup, we investigated its activation throughout the reprogramming process in microfluidics.

First, HGF and MET were differentially expressed by SR (higher HGF) and DR (higher MET) clusters and came up as early interactors in a cluster-based interaction analysis (circles and arrows in Fig. 6b). Furthermore, STAT3 nuclear target transcriptional enrichment[45] revealed their activation from day 5, along the reprogramming route (green dotted cells in Fig. 6b), in agreement with MET signalling activity. Finally, at the protein level, we observed STAT3 nuclear localization (indicative of STAT3 activation) during intermediate days (D4, D7) and at the end of the process (D12 - Supplementary Fig. 6A).

To give further evidence, we then investigated the localization of MET and STAT3 at day 6. We found that cells of smaller size (undergoing the mesenchymal-to-epithelial transition) show the highest intensity of both c-MET and nuclear STAT3 (Fig. 6c). Finally, we separately inhibited two kinases along the STAT3 axis, MET and JAK1, using small molecules and assessed reprogramming efficiency by immunostaining analysis of NANOG at day 12. Consistent with our hypothesis, we observed a significant loss of reprogramming efficiency upon inhibition of STAT3 (Fig. 6d). These data were confirmed by a direct knock-down of STAT3 mRNA using specific siRNA, that efficiently reduced both STAT3 nuclear localization in all cells at day 6 (Supplementary Fig. 6B) and reprogramming efficiency at day 12 (Fig. 6e).

Lastly, we tested whether the addition of signalling molecules was capable of further improving reprogramming yield in conventional culture systems that are otherwise far less efficient than microfluidic systems (Fig. 1a). For this purpose, we selected molecules that were found dynamically in the secretome analysis or involved in cell-cell interactions (e.g. HGF, IL6 and NRG1). In conventional culture (i.e. Petri dishes), we saw a significant increase of about 2-fold in reprogramming efficiency in terms of relative TRA-1-60⁺/NANOG⁺ area when medium was supplemented with either HGF, IL6 and its soluble receptor (sIL6R) to activate STAT3 signaling[16], and NRG1 throughout the reprogramming process (Fig. 6f). Consistent with the idea that multiple signals are involved in the first phase of reprogramming and the second phase of hiPSCs stabilization, secretome and single-cell RNA sequencing data showed more accumulation of HGF and IL6 in early phases of the reprogramming process, while NRG1 came out at later stages. To mimic this timing, we added HGF alone or with IL6/sIL6R in the first half, and NRG1 in the second half. This resulted in a further increase in the reprogramming efficiency up to three folds (Fig. 6g). However, when supplementing the medium with HGF, IL-6, sIL-6R and NRG1 together, we were able to reach the highest efficiency (i.e. 5-fold over controls), thus suggesting that the combination of specific signalling pathways further boosts hiPSCs formation (Fig. 6g).

## Discussion

Our integrative approach of secretome and single-cell transcriptomic analyses revealed a previously unappreciated crosstalk between subpopulations during the intermediate stages of human reprogramming. Whilst population heterogeneity was also described in recent papers, both in mouse[9–11] and human[4,7,8], these works reported the formation of distinctive cell clusters and diversification of pluripotent trajectories, viewing the unproductive/refractory subpopulations as a "problem" or limitation in the process. Instead, here we highlight the crucial role of reprogramming intermediates and the positive contribution of non-pluripotent clusters as actively supporting and shaping the route of the reprogramming cells towards a hiPSC identity.

The efficiency of human somatic cell reprogramming heavily relies on the successful transient accessibility and overcoming of specific intermediate stages but, given the generally low reprogramming efficiency, these stages have been hard to identify. Few strategies were previously adopted to capture human intermediate reprogramming-committed subpopulations such as cell sorting[3,4] and secondary reprogramming systems[2].

Supported by the microfluidic culture system, we took a step further through the unbiased identification of the reprogramming subpopulation trajectories and interactions based on an integrative secreted proteome and scRNA-seq analysis. The former identified a number of secreted cytokines, growth factors and ECM-related proteins actually present in the extracellular space during reprogramming and contributing to establish an environmental signaling resembling the early embryo basal lamina. scRNA-seq identified two main trajectories during reprogramming, with one almost exclusively responsible for secretory activity and one committed to reprogram. It was probably the reduced secretory activity of nascent hiPSCs or their low abundance that led previous works to overlook the role of the extracellular environment, failing to recognize nascent hiPSCs as a secretome target[4]. Recently, a few works suggested the potential for cross-population signalling in mouse reprogramming[9–11] including the role of SASP and senescence[33], but until now the molecular mechanisms and rationale behind human non-cell autonomous signalling remained unclear.

In this study, scRNA-seq could identify the putative subpopulation interaction dynamics during microfluidic reprogramming. In particular, the identification of the two distinctive trajectories, somatic secretory and reprogramming, was instrumental for scoring the putative ligand-receptor association responsible for the unidirectional support of the developmental trajectory towards pluripotency. Secretome analysis, performed here for the first time, could further reduce the dimensionality of the interactions, restricting them to those whose soluble ligand was actually detected as secreted at protein level. Only four ligands passed these restrictive selection criteria: INHBA, SPP1, NRG1 and HGF. INHBA was previously described[40], SPP1 is downstream of the HGF pathway[42], thus we focused on NRG1 and HGF, not previously implicated as reprogramming regulators. Interestingly, NRG1 signalling occurred within the reprogramming trajectory, while HGF involved population cross-talk from the secretory somatic to the reprogramming trajectory.

HGF is part of SASP, however it was not measured in Mosteiro et al., 2016[33] who instead identified IL6 in mouse cell reprogramming. Both HGF and IL6 signaling have STAT3 as a common effector, although via different receptors[46], and other works reported a positive correlation between STAT3 activity and in vivo reprogramming efficiency[16,47]. In our human reprogramming systems[2,19], IL6 was present both at transcriptional and proteomic level, however we could not detect its receptor, IL6R, in any subpopulation at any stage. Indeed, we were able to enhance reprogramming efficiency with IL6 only upon providing a soluble form of IL6R. The axis HGF/MET/STAT3 was first reported in cancer stemness and promotes the expression of pluripotent genes[46]. HGF-MET was demonstrated to take part in a mesenchymal-epithelial cross-talk[48].

HGF/MET physiological expression during development starts in the primitive streak where they take part into the so-called branching morphogenesis[49,50]. Therefore, it is intriguing to observe in our data the recapitulation of ECM organization resembling this state[51,52], with HGF secreted within the somatic trajectory, while its receptor, MET, especially present along the reprogramming one (Supplementary Fig. 6C, D).

We performed extensive experimental validation both in microfluidics and in conventional culture systems. Our loss of function data clearly show that MET activation and STAT3 signalling play an important role in preserving the efficiency of reprogramming, supporting the idea that HGF/MET/STAT3 may have a crucial role in the phenotypic conversion of developmental subpopulation towards pluripotency. Our gain of function experiments within the conventional culture system (i.e., Petri dish) support our hypothesis of the role of miniaturization in concentrating endogenous HGF and show the possibility of scaling up our findings for wider applicability. Whilst a positive role of STAT3 signalling has been extensively characterised during maintenance and induction of mouse naive pluripotency[53], STAT3 signalling pathway is not active in primed human hiPSCs. It is

therefore particularly striking that we find transient STAT3 activity to be of benefit during human reprogramming to primed hiPSC identity, and highlights that we must consider the environmental niche requirements of the intermediate states, which may differ from those of the endpoint target identity.

In our work, we followed an unbiased approach that supports the idea that the route to pluripotency can be broadened by cell-non-autonomous mechanisms. Paracrine signalling is established by highly regulated dynamics with multi-factorial contribution. We showed the use of HGF for gain of function during reprogramming in a conventional culture system, but this efficiency was amenable to further enhancement when multifactorial contributions were used. In particular, we used IL6 and soluble IL6R for a more effective downstream activation of STAT3. Moreover, we found that NRG1 contribute to enhance efficiency of hiPSC formation consistently with previous works, which upon binding ERBB2/ERBB3 receptors activates MAPK/ERK pathway and showed improved maintenance and passage of hiPSCs[54,55].

In conclusion, this work reports an overview of the environment-mediated subpopulation cross-talk during reprogramming and identifies some specific critical players. Important implications of our work are related to in vivo reprogramming, where environmental factors cannot be controlled but may affect potential applications. Moreover, strategies to reprogram in vitro fibroblasts from any donor with high efficiency are down the road and unlock the possibilities of using hiPSC as modeling systems for a large number of patients, including their use as diagnostic tools in predicting patient-specific genotype-phenotype associations in disease.

## Methods

### Microfluidic device
In this work, we used a microfluidic device, fabricated by soft lithography technique and replica molding, previously published by our group[19]. Polydimethylsiloxane (PDMS) with a 10:1 base/curing agent ratio (Dow Corning) was coupled to a borosilicate glass slide (Menzel−Gläser) through plasma treatment of surfaces.

Briefly, the microfluidic platform consists of 5 independent culture chambers, with the following dimensions: 18.8 mm of length, 1.5 mm of width, and 0.2 mm height with a 5.6 μL volume. The device is sterilized by autoclaving before use. During experiments the microfluidic chips are placed in a dish, surrounded by a water bath to reduce medium evaporation.

### Cell culture
BJ cells (Miltenyi Biotec, 130-096-726), human newborn skin fibroblasts, were cultured with complete Dulbecco's modified Eagle's medium (DMEM, Thermo Fisher, 41965 or 11965), supplemented with 10% fetal bovine serum (FBS, Thermo Fisher, 10270106 or 10099-141). Cells were maintained at 37 °C in the presence of 5% $CO_2$ and periodically tested for mycoplasma contamination.

### Reprogramming in microfluidics
Microfluidic cell cultures were performed as follows. On day 0 human fibroblasts were seeded in the microfluidic chambers, at a density of 60 cell/mm[2], after a coating with 25 μg/mL of cold fibronectin (Sigma Aldrich). Before placing chips in the incubator, 1 ml of PBS 1× was added to the bottom of the dish, in order to maintain proper humidity. From day 1 to day 8, in the morning medium was replaced using Reprogramming Medium, whereas in the night mmRNAs transfection was performed, as reported in Gagliano et al., 2019[19]. From day 9 to day 15, medium change was performed every 12 h using Pluripotency Medium.

### Reprogramming of human fibroblasts to hiPSC colonies
We generated hiPSCs from human foreskin BJ fibroblasts using microfluidic technology as previously described[19]. For proteomic analysis, a total of 10 mRNA transfections were performed using StemRNA-NM reprogramming kit (Stemgent, 00-0076) and StemMACS mRNA transfection kit (Miltenyi, 130-104-463), in E7 medium, made from E6 medium (Thermo Fisher, A1516401) supplemented with 100 ng/mL FGF2 (Peprotech, 100-18B-1000), switched to E8 medium (Stem Cell Technologies, 05990) from day 11. Whereas, for single-cell RNA-seq, 8 mRNA transfections were performed in supplemented Pluriton medium (Stemgent, 00-0070), switched to StemMACS iPS-Brew XF medium (Miltenyi Biotec, 130-104-368) from day 9. Validation experiments were performed either in microfluidics according to single-cell RNA-seq protocol or in standard 24-well plates according to manufacturer's instructions; they were performed under suboptimal conditions to enhance reprogramming efficiency differences, and medium was supplemented with HGF 100 ng/mL (Peprotech, 100-39), IL-6 50 ng/mL (Peprotech, 200-06), IL-6r 10 ng/mL (Peprotech, 200-06 R), NRG1 100 ng/mL (R&D, 396-HB), during the whole process duration, according to the specified perturbation conditions using both Pluriton medium and Nutristem hPSC XF Medium (Biological Industries, 06-5100-01-1 A) supplemented with 20 ng/mL FGF2. The loss of function experiments were performed in microfluidics supplementing the medium with Jak Inhibitor I 1uL (Millipore, 420097) and c-METi 600 uM (Selleck, PF-02341066) from day 1 to day 6. In STAT3 knock-out experiments, siRNA STAT3 10 uM (Qiagen, 1027416) or MOCK siRNA 10 uM (Qiagen, 1027284) was added in the transfection mix from day 1 to day 6. In all cases, the whole process was performed in a hypoxia incubator (5% $O_2$, 5% $CO_2$) at 37 °C.

### Sample preparation for LC-MS/MS
During reprogramming, at every medium change or reprogramming transfection, medium was collected in three replicates, pooling together the conditioned medium from the same 40 channels for each replicate. The media were stored at −80 °C until prepared for proteomic analysis. After thawing, media from four collections (two consecutive days) were pooled together. For example, sample D1-D2 was conditioned by the cells within the microfluidic chamber from day 1 to day 3 mornings. 3 kDa cut-off centrifugation membranes (Amicon Ultra 0.5 mL, Ultracel 3 K, Merck) were used for filter-aided sample preparation (FASP)[56]. Proteins were concentrated by centrifugation for 20 min at 4 °C and 14,000 g, then washed twice with a 50 mM triethylammonium bicarbonate (TEAB, Thermo Scientific) buffer containing 8 M urea (Sigma-Aldrich). Protein content was quantified by Pierce BCA Protein Assay Kit (Thermo Scientific). Each sample proteins were reduced for 60 min at 56 °C with 100 mM DTT (Sigma-Aldrich), and alkylated for 30 min at room temperature in the dark with 55 mM iodoacetamide (Sigma-Aldrich). Samples were washed with 50 mM TEAB for three times. An equal amount of protein for each sample was digested by trypsin (Promega) at 37 °C for 16 h. Digested peptides were desalted by C-18 spin column (Pierce) and vacuum dried. Then, labeling by 6-plex Tandem Mass Tag (TMT6, ThermoScientific)[28] was performed according to manufacturer's instructions using 50 μg of peptides from each sample. The six-time point samples of each of the three replicates were pooled, then desalted and vacuum dried.

### Mass spectrometry analysis
25 pre-fractions were collected on UPLC (Agilent 1290) with high pH C18 column (2.1 mm × 30 mm). Before MS analysis, peptides were resuspended in 10 μL of 0.1% formic acid. Thermo Fusion Mass Spectrometer coupled with Thermo EasynLC1000 Liquid Chromatography was used to get peptides profiles. 90 min of LC-MS gradients were generated by mixing buffer A (0.1% formic acid in water) with buffer B (0.1% formic acid in 80% ACN in water) by different proportions. Using NSI as the ion source and Orbitrap as the detector, the mass scan Range was at 300-1800 m/z, and the resolution was set to 120 K. The MS/MS was isolated by Quadrupole and detected by Ion trap, whose resolution was set to 60 K. The activation type was HCD.

## Proteomic bioinformatic analysis

Peak list files were searched against UniProt human reference proteome (UP000005640) by MaxQuant (v. 1.6.3.4)[57]. TMT6 modification and carbamidomethyl on cysteine were set as fixed modifications. The oxidation of methionine, acetylation of protein N-terminus, and phosphorylation (STY) were set as variable modifications. Peptide-spectrum matches (PSMs) were adjusted to 1% and then assembled further to a final protein-level false discovery rate (FDR) of 1%. Proteins not identified in at least 2 replicates in at least one time point were excluded from further analysis. Common contaminants (keratins and Bos taurus proteins) were also filtered out, for a final number of 4542 proteins identified. Missing values were imputed by the mean value of the other two replicates. TMT intensities were normalized according to BCA quantification to obtain a relative quantification proportional to protein concentration in culture. The distributions of the three replicates of TMT intensities were scaled by their respective medians. A principal component analysis (PCA) was performed in MATLAB R2017a (The Mathworks) using mean-centered TMT intensities. A list of secreted proteins was manually annotated by integrating the following resources: secreted proteins predicted by MDSEC as reported in Protein Atlas database[58] (http://www.proteinatlas.org), secreted proteins from Data S1 in Gonzalez et al., 2010[59]; a list of ligands from Gene Ontology-Molecular Function categories "cytokine activity", "growth factor activity", and "hormone activity", and senescence-associated secreted proteins (SASP) annotated from literature[60–63]. Of the proteins identified in this study, only those secreted according to the criteria above were further studied, in order to avoid the proteins possibly derived from cell death. Differentially secreted proteins between time pairs were assessed with student t-test, using a threshold of 5%. Proteins whose concentration was maximal only at the first time point (D1-D2 sample) were excluded from further analysis, as potential residual proteins from FBS used during fibroblast expansion. Functional enrichment analysis of Reactome pathways was performed using ReactomePA (v1.36.0)[64] Bioconductor package. Reactome hierarchy was visualized using ClueGO (v2.5.6)[65] within Cytoscape (v3.8.0)[66]. Genes specific to different human embryonic stages were derived from a published single-cell RNA-seq study[29], of these core ECM genes were selected based on the annotations in Naba et al., 2012[36]. Proteins playing a role as ligands were taken from Ramilowski et al., 2015[39]. Hierarchical clustering with heat map data visualization was performed in MATLAB R2017a, using Euclidean distance and complete linkage.

## Sample preparation for single-cell RNA-seq

For each time-point, cells were detached using TrypLE-express (ThermoFisher, Gibco 12604). Harvested cells were then centrifuged at 300 g and resuspended at the final cell density of 100 cells/mL using a solution containing 40% KnockOut Serum Replacement (KSR, ThermoFisher, Gibco 10828) in DMEM. For each timepoint, two replicates were produced, each containing cells from 4 independent chips that were pooled together then divided in aliquots containing 5000-80,000 cells. Samples were cryopreserved in DMEM supplemented with 40% KSR and 15% DMSO and stored in liquid nitrogen.

scRNA-seq libraries were generated using one or two samples for each replicate. Briefly, each cryopreserved aliquot was thawed at 37 °C until a tiny ice crystal remained in solution. Then each sample was diluted under gentle shaking by dropwise adding 10 volumes of DMEM supplemented with 40% KSR. Cells were washed twice using a washing buffer containing 8% MACS Running Buffer (Miltenyi, 130-091-221) in PBS. Cells were then resuspended in the washing buffer and filtered through a 40 μm cell strainer (Biosigma, 010198Z). Cell viability and concentration were checked by visual inspection using Trypan Blue (Logos Biosystems, L12002).

Single-cell RNA seq libraries were produced according to 10X Single Cell 3' v2.0 standard protocol and sequenced on Novaseq 6000 (Illumina).

## Single-cell RNA-seq data pre-processing

scRNA-seq data pre-processing was performed using the cellranger software (v 2.2). Fastq files were generated using the Cellranger pipeline *mkfastq* using 10X standard Chromium barcode sequences. Alignment, filtering, barcode and UMI counting were performed using the Cellranger *count* pipeline. Human pre-built genome index has been applied (hg38 genome reference and GRCh38 annotation, including protein coding, linc and antisense RNAs). Each feature-barcode matrix from each independent sample was merged to build up the final dataset, containing 33,694 genes and 44,197 cells, then subjected to cells and genes filtering. Cells having less than 1000 detected genes and with the mitochondrial associated reads percentage greater than 10% were filtered out. Furthermore, in order to have a homogenous sampling for each reprogramming day, the cell dataset was randomly subsampled to 2500 cells per time point. The final dataset retained only those genes expressed in at least 5% of all the cells, leading to 12,932 total genes. Gene expression values were normalized to CPM (counts per million) and transformed to the $\log_2$ scale using a pseudocount of 1. Finally, cell-cycle scores and, consequently, phases were assigned to each cell by Seurat's (v.3.1.5) CellCycleScoring function.

## Single-cell RNA-seq data visualization and clustering

To better visualize and characterize single cell data, high dimensionality was reduced. First, we computed the neighborhood graph using the function *compute_neighborhood_graph* from the Python (v 3.9.5) package wot (v 1.0.5)[10], using 50 neighbors and choosing the first 100 PCA components and the first 20 diffusion map components. The resulting 120 components were used as input to initialize the Force-Directed Layout Embedding (FLE) algorithm, using forceatlas2 (v 1.0.3) with 1000 iterations and reducing the space to 2 dimensions (FLE1 - FLE2). The same components were also applied to perform an unsupervised graph-based algorithm (louvain) using the FindNeighbours and FindClusters (resolution = 0.6) functions in the Seurat (v.3.1.5)[67] package. This step resulted in the identification of 12 clusters, annotated based on the enrichment of somatic and developmental signatures[2] at the single-cell level (SR = somatic related; DR = developmental related; NA = not assigned) and ordered by their composition in terms of time-points.

## Single-cell RNA-seq differential gene expression and gene sets enrichment

Differentially expressed genes among clusters were identified using the FindAllMarkers function from Seurat (v.3.1.5), taking just LFC ($\log_2$ fold change) more than 0.25. For each gene, significance was assessed with the Wilcoxon rank-sum test *P* values, adjusted for multiple testing using the Benjamini–Hochberg correction to retrieve the false discovery rate (FDR). Only genes with FDR < 0.01 were considered. As expected, many gene markers were shared by clusters from the same group (SR or DR) because of the continuous nature of data. We therefore decided to select unique markers and to take duplicated markers once, preferring the cluster where the LFC was the highest.

To perform enrichment of gene signatures in clusters, we used pre-ranked Gene Set Enrichment Analysis (GSEA) from fgsea (v 1.14.0)[68] R package. Pre-ranked lists for each cluster were generated by assigning to each gene its LFC relative to the average expression across all the other clusters. Common pathways were defined as belonging to several databases, i.e. Hallmark[69], KEGG, Biocarta, Reactome and Gene Ontology Biological Process.

Enrichment scores (ES) of gene signatures at the single cell level were obtained by computing the z-score for each gene across the data

sheet. After truncating these scores at 5 or −5, the enrichment score was defined by the average z-score over all genes in the gene set.

## Single-cell RNA-seq trajectory inference

To infer the reprogramming trajectory, two different approaches were used: wot (v 1.0.5)[10] and Monocle3 (v 0.2.3.0)[38]. The former applies the Mass Optimal Transport theory to the gene expression space to infer, for each cell in a given sample, the most probable ascending and descending cells in the previous and following timepoints. First, birth-death rates were computed for each cell by applying a logistic function to the enrichment scores for Cell-cycle[70] and Apoptosis (R-HSA-109581, hsa04210, HALLMARK_APOPTOSIS in Liberzon et al., 2015[69]). ß and δ logistic functions were optimized (center = −0.1 and center = 0.15, respectively). Second, transport maps were generated in batch for each pair of subsequent time-points using the functions *wot.o-t.OTModel* (epsilon = 0.2) and *compute_all_transport_maps*. Finally, trajectories were inferred using *population_from_cell_sets* and *trajectories* functions starting from D15 cells that showed high enrichment (> 2) for the signatures Matrisome[36] and Late pluripotency[2]. For each timepoint, cells having a trajectory probability greater than the mean were considered to belong to the trajectory.

Monocle 3, on the other hand, learns a trajectory graph looking at the gene expression changes required for each cell to move from a state to another during a dynamic biological process. In particular, UMAP coordinates in Monocle 3 were replaced with the FLE ones, in order to obtain an FLE-based Monocle trajectory. Furthermore, *cluster_cells* and *learn_graph* were performed by tuning the parameters *k* (30) and *ncenter* (96), respectively.

## Single-cell RNA-seq interaction analyses

Interaction analyses have been performed on a set of 82 ligand-receptor pairs obtained as follows.

A putative list of 3333 couples has been generated from the ligands identified in the secretome analysis with every possible receptor. Afterwards, receptors have been filtered out in case they were not defined as receptor on BioGrid or they did not belong to any of these GO terms: GO-CC:0009897, GO-CC:0098802 and GO:0004714. The resulting list of 1082 pairs was then filtered based on the expression of both ligand and receptor in at least one cell (491). Finally, we selected only those pairs that were experimentally validated[39].

Interaction scores between trajectories throughout the time-course were evaluated as shown in Schiebinger et al., 2019[10] (Approach 1). Top interactors were selected by ordering the results by standardized interaction score (sIS). Then, the highest ligand-receptor pair for each day was assessed. All the unique couples with a sIS comprised between the first and the last day-specific occurrence was taken.

HGF/MET cluster-to-cluster interaction scores were computed as the product between the average gene expression value of MET in a cluster and the value of HGF in another. Significance was assessed with empirical *p*-value, generating a null distribution of 1000 permutations on the association between cells and clusters.

Matrisome to late-pluripotency interaction was also evaluated using a different, independent approach (Approach 2) using scSeqComm[71]. First, each ligand and each receptor in a trajectory was scored based on the probability of observing expression values higher than the ones observable by chance from the expression levels of random genes in the same trajectory and time point. Second, ligand-receptor pairs scores were computed as the minimum (i.e. a fuzzy logical AND operator) between the ligand score and the receptor score. An empirical *p*-value was also computed doing the above procedure multiple times on a randomly permuted gene expression level matrix (i.e. permuting multiple times the gene expression levels of each cell independently) and then measuring the percentage of interaction sub-scores higher than the obtained one. The score and the

corresponding *p*-value was computed as a function of both the absolute ligand and receptor expression levels (as explained above) and, similarly, for their $\log_2$ fold change in a specific trajectory/time point, with respect to their average expression across the entire data matrix.

## STAT3 targets expression

STAT3 targets were identified using a ChIP-seq dataset on HUS64 human embryonic stem cells[45]. In particular, STAT3 target genes were defined as genes with STAT3 significant peaks at ±3000 bp from the transcription start site. For each cell, the STAT3 pathway enrichment was computed from the scaled gene expression matrix as the average value for all the STAT3 targets. For each enrichment value, the corresponding *p*-value was calculated by performing a hypergeometric test and using a random gene list to obtain the null distribution.

## Bulk RNA-seq analysis of reprogramming data

To analyze the relationship between mouse feeders and human reprogramming cells at day 8, we re-analyzed bulk RNA-seq data from Cacchiarelli et al., 2015[2]. Fastqs have been trimmed using Trim Galore (https://github.com/FelixKrueger/TrimGalore) for quality and adapters removal. Then, reads have been mapped with TopHat (v. 2.1.0)[72] and Bowtie2 (v. 2.3.2)[73] with default parameters against a hybrid build of the human (hg38) and mouse (mm10) genomes. Reads aligned to the mouse reference were few (alignment rate <20%), but it was consistent with the purified nature of the samples, where mouse cells should just represent contamination. Finally, read quantification was performed with HTSeq (v. 0.9.1)[74] on GENCODE human (GRCh38) and mouse (mm10) genome annotations, including protein coding, linc and antisense RNAs. The final count matrix was created by merging mouse and human genes by orthology and differential expression analysis was performed between human and mouse (feeders) samples using DESeq2[75].

## Immunofluorescence staining

For immunofluorescence staining, cells were fixed in 4% paraformaldehyde for 10 min at room temperature, then permeabilized with 0.1% Triton X-100 for 10 min, blocked in blocking solution (DPBS with 10% horse serum and 0.1% Triton X-100 for intracellular targets) for 45 min, followed by overnight incubation with primary antibodies. The following antibodies were used for immunofluorescence: rabbit anti-NANOG (Cell Signaling, 4903)(1:200), mouse anti TRA1-60 (Millipore, MAB4360)(1:100), mouse anti-STAT3 (Cell Signaling, 9139) (1:300), goat anti- HGFR/c-MET (R&D, AF276)(1:200). Alexa488 or Alexa594 conjugated rabbit, mouse or goat secondary antibodies (1:200) were used (Life Technologies, A21202; A21207; A11058). The nuclei were stained with Hoechst 33342 (Life Technologies).

Images were acquired on a confocal TCS SP5 microscope (Leica) at 40x magnification and on a fluorescence microscope DM6B (Leica) at 5 and 10x magnification.

## Assessment of reprogramming efficiency

Reprogramming efficiency was quantified after immunostaining with TRA1-60 and NANOG markers. When the efficiency of reprogramming was too high to allow counting single colonies, it was quantified as relative TRA1-60+ and NANOG+ cell area divided by the total area occupied by the cells. Since TRA1-60 is a membrane/extracellular marker and NANOG is a nuclear marker, we considered TRA1-60 area positive only where it overlapped with NANOG positive nuclear area, for having the double positive cells as result.

## Microarray data analysis

Previously published microarray data[18] were analyzed by the Quantitative Set Analysis for Gene Expression (QuSAGE v2.26.0)[76] Bioconductor package within MSig DB – Hallmark gene set collection[69]. Results were plotted by MATLAB R2017a.

## Article

## Secondary reprogramming experiments

Secondary reprogramming experiments were performed as previously reported in Cacchiarelli et al., 2015[2]. Briefly, $10^5$ TERT-immortalized secondary fibroblasts (hiF-T) harbouring a doxycycline-inducible OSKM cassette were seeded with or without irradiated mouse embryonic fibroblast (MEF) in a 3:1 ratio. The day after seeding, cells were treated with doxycycline (Sigma Aldrich, D9891-1G) (2 μg/mL) to start the OSKM expression. In addition, LSD1 inhibitor RN-1 (MERK, 489479) was added at the final concentration of 10 nM to further increase the reprogramming efficiency. Both treatments were prolonged for 21 days. Colony counting and visualization in bright-field were performed by using a TRA-1-60 chromogenic staining[77].

## Statistics and reproducibility

Sequencing data were analyzed and plots were produced in R[78] (v 4.2.0). Data variability is presented as boxplots, where bars indicate the median, boxes indicate the 25th and 75th percentiles, whiskers represent median +/- the interquartile (25-75%) range multiplied by 1.5. The number of replicates and the tests used to assess statistical differences are reported within each figure caption. Experiments shown in Fig. 1C have been repeated 10 times independently with similar results.

## Reporting summary

Further information on research design is available in the Nature Portfolio Reporting Summary linked to this article.

## Data availability

The scRNA-seq raw and processed data generated in this study have been deposited in the GEO database under accession code "GSE221739". The proteomic data generated in this study have been deposited in the Massive database under accession code "MSV000090954 [https://doi.org/10.25345/C5M32NG9C]". Other databases enquired are: Protein Atlas, Gene Ontology, Reactome and MSig DB. All other relevant data supporting the key findings of this study are available within the article and its Supplementary Information files or from the corresponding author upon reasonable request. Source data are provided with this paper.

## Code availability

The authors declare that data analysis in this study was performed with bioinformatic algorithms already publicly available. However, we provide R scripts used to analyze scRNA-seq data. (DOI: 10.5281/zenodo.7640602; https://github.com/panariellofrancesco/scRNAseq_Reprogramming)[79].

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

## Acknowledgements

This research was supported by the European Research Council (grant agreement 101055300 - ReprOids to N.E.), the Telethon Foundation Award (GGP15275 and GGP20105 to N.E.), and STARS Wild Card Grants (STARS-WiC award to N.E.) and Cariparo Foundation. This work was supported by Fondazione Telethon Core Grant, Armenise-Harvard Foundation Career Development Award, European Research Council (grant agreement 759154, CellKarma), and the Rita-Levi Montalcini program from MIUR (to D.C.). Ca.L., S.M., M.H., M.C. and N.E. were supported by grant F-0301-15-009 by ShanghaiTech University. Ca.L. was supported by the Natural Science Foundation of China (31601178). This work was partially funded by PROACTIVE 2017 "From Single-Cell to Multi-Cells Information Systems Analysis" (C92F17003530005, Department of Information Engineering, University of Padova) and "Research Grant (type B) – B senior initiative" (Department of Information Engineering, University of Padova). We are grateful to SIAIS Analytical Platform (Dr. Wenzhang Chen and Dr. Wei Zhu) at ShanghaiTech University for mass spectrometry analyses.

## Author contributions

O.G., F.P., Ca.L., D.C. and N.E. designed the study. Ca.L. and N.E. designed the proteomic experiments. W.Q. and M.H. performed microfluidic platform production for proteomic experiments. Ca.L. performed reprogramming for proteomic analyses, bioinformatic analysis on proteomic and microarray data. Ca.L., S.M., and M.C. performed proteomic sample preparation. O.G. designed and optimized microfluidic reprogramming and performed microfluidic reprogramming experiments for scRNA-seq analysis and for LoF and GoF. S.A. and W.Q. helped in reprogramming experiments. A.G. and A.M. pre-processed samples for scRNA-seq and performed library preparation. F.P., A.G. and D.C. designed the sc-RNA-seq experiments. F.P. analyzed scRNA-seq data. F.P. and G.S. performed trajectory inference analysis. O.G. and S.A. performed immunofluorescence stainings, image acquisition and analysis. P.A., L.V., S.S. and M.S. helped in analyzing bulk RNA-seq data. S.R. and M.D. sequenced scRNA-seq libraries. V.B. helped in sample management. Ce.L. helped image acquisition and analysis. H.S. helped in the design of the loss of function experiments. G.B. and B.D.C. performed interaction score analyses. F.P. designed and prepared the figures. O.G., F.P, Ca.L., A.G., D.C. and N.E. critically discussed the data and wrote the manuscript. D.C. and N.E. supervised the project.

## Competing interests

O.G, Ca.L. and N.E. are co-inventors on patent applications describing the reprogramming and differentiation processes in microfluidics, application number PD2013A000220, IT UA20162645 and 102016000039189 and PCT/IB2017/052167. O.G. and N.E. are co-founders of Onyel Biotech Srl. D.C. is founder, shareholder, and consultant of Next Generation Diagnostic srl. The remaining authors declare no competing interests.
