## [Peer Review File · Nature Communications]

REVIEWER COMMENTS

Reviewer #1 (Remarks to the Author):

The authors used LC-MS/MS to quantify proteins in conditioned media at 6 time points during human reprogramming. They focused on 555 secreted proteins and found that signaling pathways such as ECM gradually change during the process. Combined with scRNA-seq data, the authors found that most of the secreted proteins were derived from the Somatic-Related clusters (SR). The authors found that HGF and NGR ligands secreted by the SR subpopulations act on Developmental-Related clusters (DR) and promote reprogramming in the middle and late stages of reprogramming. This study reveals the phenomenon of synergistic reprogramming of multiple cells, which is informative for resolving the intermediate state of reprogramming and further improving reprogramming efficiency. Although this study is interesting, some questions require further elaboration by the authors.

Major issues:

1. The proteins shown in the figures all appear to increase with the reprogramming process, are there decreasing proteins? If so, what pathways do these proteins correspond to?
2. While SR2 has a positive effect on reprogramming, in turn, is the generation of SR2 associated with DRs? Or is it related to other SR subpopulations?
3. Do SR and DR subpopulations in single-cell data correspond to the cell types of embryonic development?
4. There is evidence that different reprogramming factors may drive the generation of different reprogramming branches (PMID: 30772174, 31548608). The authors should perform transcriptional regulation analysis to investigate whether the expression of NGR and HGF is regulated by reprogramming factors.

Minor issues:

Are the median of detected genes and UMI counts of NA cluster in Figure 2C comparable to other clusters?

Authors should double-check reference citations; for example, in line 308, reference 8 should correspond to the mouse study, not the human study.

There are writing mistakes in the figure 3 legend, for example, Liu et al., 20207, Liu et al., 20207.

Reviewer #2 (Remarks to the Author):

The manuscript by Gagliano et al. reports on the role of the secreted factors in human pluripotent cell reprogramming. The manuscript is clearly written and organized and provides a novel insight into the role of distinct cell populations in cell reprogramming by providing secreted signaling molecules. As such, I can recommend it for publication.

There are some minor questions and need for clarifications/corrections:

1. As NRG1 gene can produce several isoforms, could authors clarify what isoform was identified in the cell secretome (perhaps what peptides specifically were identified). In cell culture supplementation experiments, NRG1-b1/HRG1-b1 was used. In the methods section, product number is specified, but again it might be clearer if the specific isoform is directly identified in the manuscript (both results and

methods).

2. Methods line 402 - 422 Why were experiments for proteomic secretome analysis performed in different media than sc-RNAseq experiments?

3. Although reference for detailed reprogramming protocol is provided, some details should be clearly stated here - what was the volume of medium in microfluidics chamber collected for experiments, how often was the medium changed (from description of Sample preparation for LC-MS/MS, four collections over 2 consecutive days suggest medium was changed twice a day, but the referenced protocol states as an improvement once a day medium change).

4. Methods line 404 - corresponding reference number for Gagliano et al 2019 missing.

5. Methods lines 620-622, Immunofluorescence staining: goat anti-HGFR/c-MET antibody is mentioned, but no corresponding anti-goat fluorescence conjugate is provided.

6. Methods lines 626-630: Reprogramming efficiency is described as combined TRA-160 and NANOG positive area, but in figures from microfluidics experiments, only NANOG staining is used. This should be clarified.

Code and data availability statements (lines 678 - 685):

The datasets from proteomic and RNA-seq are stated to be deposited before publication, but no link to the datasets is provided in the manuscript. This should be provided to reviewers.

For code, authors state use of publicly available algorithms, but it would help assessing analytical steps taken if the scripts/code used for this particular data analysis were provided as a supplementary material.

Figures:

Fig2B - it would be better to use color palette similar to 2C, to allow clear identification of cell populations from collected time-points. Gradual color palette used doesn't allow simple identification. This would also allow easier comparison with identified clusters in 2C.

Figure 3J - Liu et al., 2020⁷ should probably be just Liu et al., 2020⁷ similarly in K), 2015² - should this be 2015²? Probably reference number - should be upper index.

Figure 4C - are there some labels overlaid over cells? With this size of the image, it is hard to distinguish, but there seem to be some artifacts visible in the middle of each cell. As these don't seem to serve here any purpose, it would be better to display original IF image without labels.

Figure 4E - Anova with post-hoc Tukey test is indicated for this graph, although it compares only 2 conditions - was it really the test used?

Figure 4F,G - why was single color representation used for double immunostaining of NANOG/TRA-1-60? Only NANOG was displayed for cells reprogrammed in microfluidics device, it might be probably more useful to display only NANOG here as well.

Figure 4G - should there be "temporally modulated"?

Reviewer #3 (Remarks to the Author):

In this manuscript, the Authors exploit a microfluidic platform that they have spent years refining, and apply it to the definition of the paths chosen by cells towards human pluripotency. The Authors argue that this device allows them to define intercellular factors contributing to the generation of pluripotency, in line with what has been already shown in mouse cells.

On the overall, I think the approach and the findings are intriguing. The suggestion that apparently "dead end" populations may instead be conducive of the reprogramming is very interesting. I feel however that a few key aspects should be properly addressed, before recommending publication in Nature Communication. Specifically, I understand that the microfluidic approach is very useful in increasing the yield of the generated iPS, based also on the previous publications of the group, including the Nature Protocols papers. However, the Authors in this manuscript aim to uncover the biological underpinning of reprogramming. As such, several choices that are made, including the media and growth factors, the experimental settings and more broadly the small and big experimental choices that were made, question the physiological relevance of their findings. I think that a much more detailed methodological description should be included, and most importantly, appropriate positive and negative controls added, to assuage the concern that their findings could be artifactual, or at least driven by the chosen experimental settings. Just to provide an example, in line 89 is mentioned that "A medium that contains only 3 types of proteins (FGF2, INS, TF) preserved the high efficiency of microfluidic reprogramming". How this choice would affect the reprogramming? if some of these factors were to be replaced, or else others added or removed, would be the specific and overall conclusions of the paper maintained?

Minor points:

- a more detailed description of the microfluidic platform used should be included in the material and methods section. Also, figure 1A is not very informative. I would replace the fig1A with extended figure 1A, and with a more detailed description on the corresponding figure legend (now limited to "A) Schematic representation of the experimental design for the proteomic study."). Without reading the previous technical papers from the same authors, it is difficult to understand the methodology.
- what is the meaning of "Protein tagging allowed us to obtain a relative quantification of each protein along the process (Extended Data Fig. 1A)."? In the methods and figure legends this information should be included.
- It is unclear the statement "We quantified 4542 proteins, identified in either two (19%) or three replicates (81%)." I would imagine a higher number from 2 replicates.
- I would ask the Authors to show the data reported in "Therefore, as NRG1-ERBB3 expression occurs only along the reprogramming trajectory, we did not get significant results when comparing MEFs versus human reprogramming intermediates from our human secondary system (data not shown)", line 246.

Response point-by-point to reviewers

We thank all the reviewers for their helpful comments and appreciation of our work. In order to make our work clearer and avoid misunderstanding, in agreement with their suggestions, we split the original four figures into six new figures and we added new supplementary figures to fully address his/her comments.

Briefly:

- Original Figure 1 is now split into two figures, separating the overall project design and the secretome results: Figure 1 now shows the microfluidic details, the sampling scheme of scRNA-seq and of secretome analysis, and the overall findings from a general analysis of these data. In particular, we show representative images of the reprogramming progression and quantitative single-cell violin plots to show hallmark gene activation. Finally, we show specific trends of differential protein and RNA up-down regulation identifying boosts of transcriptional waves and protein secretion;
- Figure 2 is the bottom part of the original Figure 1;
- Figure 3 is a slightly revised version of the original figure 2;
- Figure 4 and Figure 5 are derived from a slightly revised and split original Figure 3 to efficiently describe the scRNA trajectory and the cross-talk in two separated figures;
- Figure 6 is the original Figure 4.

Reviewer #1

The authors used LC-MS/MS to quantify proteins in conditioned media at 6 time points during human reprogramming. They focused on 555 secreted proteins and found that signaling pathways such as ECM gradually change during the process. Combined with scRNA-seq data, the authors found that most of the secreted proteins were derived from the Somatic-Related clusters (SR). The authors found that HGF and NGR ligands secreted by the SR subpopulations act on Developmental-Related clusters (DR) and promote reprogramming in the middle and late stages of reprogramming. This study reveals the phenomenon of synergistic reprogramming of multiple cells, which is informative for resolving the intermediate state of reprogramming and further improving reprogramming efficiency. Although this study is interesting, some questions require further elaboration by the authors.

We thank the reviewer for his/her helpful comments and appreciation of our work.

Major issues:

1. The proteins shown in the figures all appear to increase with the reprogramming process, are there decreasing proteins? If so, what pathways do these proteins correspond to?

To perform the pathway enrichment analysis (Fig. 1D now in Fig. 2A), we filtered out only proteins with maximum value at D1-D2 to avoid potential residual proteins from FBS used during fibroblast expansion (as stated in the Methods section of the manuscript, page 10). In the heatmaps (Fig. 1E-F, now Fig. 2B-C), proteins belonging to specific categories are shown, without any selection based on the expression trend.

As the reviewer can now appreciate from the bottom right of the new Figure 1D (copied here below), if we exclude proteins decreasing from the first transitions (for the reasons stated above), no new proteins are downregulated during the time course (at later stages the downregulation mostly corresponds to the proteins upregulated in the previous stages).

2. While SR2 has a positive effect on reprogramming, in turn, is the generation of SR2 associated with DRs? Or is it related to other SR subpopulations?

We thank the reviewer for pointing out this interesting point. Based on our data, the time points that contribute to SR2 and DR clusters show some overlap on days 7, 9, and 11 (see table below). Nevertheless, the majority of cells (almost 90%) that compose the cluster SR2 are present on days 1-7. We cannot exclude that the emergence of cluster DR at day 7 increases the number of SR2, but if this happens it will be limited to the residual <10% of cells. Moreover, the trajectory of both SR and DR belong to distinctive paths (Fig. 3B, now Fig. 4B). The genes of SR2 belong to a set of genes that swiftly increase at the first time point upon adding the reprogramming factors when not even a single DR cell exists. We added a specific comment in the text to better clarify the cross-talk reciprocity between DR and SR2 (highlighted on page 5).

The table shows the number of cells belonging to each cluster (SR/DR) and days (DX)

	D0	D3	D5	D7	D9	D11	D13	D15
SR1	2373	15	3	5	0	0	0	0
SR2	73	847	702	345	167	81	2	1
SR3	1	1569	801	300	179	4	0	0
SR4	0	5	285	348	28	7	0	0
SR5	0	2	632	1143	91	19	0	0
SR6	0	0	1	32	502	592	327	112
SR7	0	0	0	2	69	411	184	98
DR1	0	0	30	313	1368	53	5	0
DR2	0	0	0	0	62	1209	393	24
DR3	0	0	0	0	3	43	1257	1907
DR4	0	0	0	0	1	57	248	232
NA	53	62	46	12	30	24	84	126

3. Do SR and DR subpopulations in single-cell data correspond to the cell types of embryonic development?

We thank the reviewer for suggesting this idea. For this purpose, we performed an enrichment analysis on the single-cell map with the embryonic signatures belonging to the different early embryo human stages (Boroviak et al., 2018 - PMID: 30413530). From this analysis here below, we can appreciate that all the cells expressing the early embryo signatures (red) belong to the right part of the map, the same leader line of cells transitioning from common to pluripotency (Figure 4B) and where STAT3 is activated (Figure 6B). We added this analysis in Extended Data Fig. 3C and added this to the text on page 4 (highlighted).

4. There is evidence that different reprogramming factors may drive the generation of different reprogramming branches (PMID: 30772174, 31548608). The authors should perform transcriptional regulation analysis to investigate whether the expression of NGR and HGF is regulated by reprogramming factors.

To this end, we analyzed publicly available ChIP seq reference data of the OSKM factors during human reprogramming (Soufi et al., 2012 - PMID: 23159369). In all of the time points analyzed we found only one peak for MYC binding the HGF promoter, details below:

ChIP	Chr	start	end	Gene	Distance from TSS
CMYC	chr7	81237158	81237358	HGF	127 bp (promoter)

We can conclude that, with the exception above, the reprogramming factors do not directly regulate HGF and NRG expression. This is in line with the fact that the reprogramming factors are also expressed in iPSC/ESC, but no HGF or NRG results are expressed in pluripotent lines.

Minor issues:

Are the median of detected genes and UMI counts of NA cluster in Figure 2C comparable to other clusters?

Based on this hint we re-evaluated our data. This analysis was a great improvement for us: we always had to exclude the NA cluster for biological inconsistencies, but in reality, it was just a cluster containing low-quality cells (we added the diagnostic plot of counted genes in Extended Data Fig. 3A).

This new analysis shows that cluster NA is qualitatively lower.

Authors should double-check reference citations; for example, in line 308, reference 8 should correspond to the mouse study, not the human study.

Well noted and updated

There are writing mistakes in the figure 3 legend, for example, Liu et al., 20207, Liu et al., 20207.

Well noted and updated

Reviewer #2

The manuscript by Gagliano et al. reports on the role of the secreted factors in human pluripotent cell reprogramming. The manuscript is clearly written and organized and provides a

novel insight into the role of distinct cell populations in cell reprogramming by providing secreted signaling molecules. As such, I can recommend it for publication.

We thank the reviewer for the great appreciation of our work!

Minor issues

1. As NRG1 gene can produce several isoforms, could authors clarify what isoform was identified in the cell secretome (perhaps what peptides specifically were identified). In cell culture supplementation experiments, NRG1-b1/HRG1-b1 was used. In the methods section, product number is specified, but again it might be clearer if the specific isoform is directly identified in the manuscript (both results and methods).

Experimentally, we identified NRG1 by LC-MS/MS by the following unique peptides, which were BLAST aligned on Uniprot to the reviewed NRG1 different isoforms:

- ASLADSGEYMCK, common to Pro-neuregulin-1 and isoforms 2, 3, 4, 6, 7, 8, 9, 11, 12
- CETSSEYSSLR, common to Pro-neuregulin-1 and isoforms 2, 3, 4, 6, 7, 8, 9, 11, 12
- DLSNPSR, common to Pro-neuregulin-1 and isoforms 2, 3, 4, 6, 7, 8, 9, 10, 11, 12
- NKPQNIK, common to Pro-neuregulin-1 and isoforms 2, 3, 4, 6, 7, 8, 9, 11, 12

As shown, these peptides do not make it possible to discriminate between different isoforms, and we picked the beta NRG isoform as the one commercially available and validated to be biologically active.

2. Methods line 402 - 422 Why were experiments for proteomic secretome analysis performed in different media than sc-RNAseq experiments?

Cell-secreted proteins in the extracellular space represent a complex mixture of many proteins accumulated at different concentrations. Microfluidics enhances the concentration of these proteins, increasing the signal-to-noise ratio compared to a conventional culture system, meaning that, because of its small volume, the ratio of cell-secreted protein concentration to medium protein concentration is higher in microfluidics. However, medium proteins still represent a relevant background noise for the measurement. Keeping medium proteins at a minimum in conditions that still promote high-efficiency reprogramming represents an optimal condition for cell-secreted protein detection. This is the reason why for the secretome study we changed the medium composition to a low protein medium.

To compensate for the reduced reprogramming efficiency induced by the use of a low protein medium, during reprogramming transfections, we also introduced LIN28 and NANOG mRNAs and pluripotent miRNAs, which we already proved to facilitate reprogramming WITHOUT altering its main trajectories (Cacchiarelli et al, 2015 - PMID: 26186193 and Zhang et al, 2016 - PMID: 27133794). This allowed for reaching a comparable reprogramming efficiency in low protein conditions.

3. Although reference for detailed reprogramming protocol is provided, some details should be clearly stated here - what was the volume of medium in microfluidics chamber collected for experiments, how often was the medium changed (from description of Sample preparation for

LC-MS/MS, four collections over 2 consecutive days suggest medium was changed twice a day, but the referenced protocol states as an improvement once a day medium change).

Thanks for this suggestion. We added more experimental details in the Material and methods section (Cell culture and Reprogramming in microfluidics paragraphs, highlighted on page 9) regarding how to perform cell seeding and mmRNAs reprogramming in microfluidics.

4. *Methods line 404 - corresponding reference number for Gagliano et al 2019 missing.*

Thanks, we corrected it as suggested.

5. *Methods lines 620-622, Immunofluorescence staining: goat anti-HGFR/c-MET antibody is mentioned, but no corresponding anti-goat fluorescence conjugate is provided.*

Thanks, we added the secondary anti-goat antibody as suggested on page 13 (highlighted).

6. *Methods lines 626-630: Reprogramming efficiency is described as combined TRA-160 and NANOG positive area, but in figures from microfluidics experiments, only NANOG staining is used. This should be clarified.*

We quantified the reprogramming efficiency as a positive area for both TRA-160 and NANOG, which means that the colonies considered pluripotent must be double positive. As can be now seen from representative new Figure 1C (left) with both NANOG and TRA-160 have a very high concordance rate among all the experiments (despite the first marker being nuclear and the second being membrane/extracellular).

As a result of this high concordance rate, in the screening of Figure 6D and 6E, we decided to use NANOG only. We understand that this could be misleading and we described this better in the methods section on page 13 (highlighted).

Code and data availability statements (lines 678 - 685):

The datasets from proteomic and RNA-seq are stated to be deposited before publication, but no link to the datasets is provided in the manuscript. This should be provided to reviewers.

There is now a reference to all the omics data on page 14. Data is being kept private until publication, thus reviewers might ask for a token to visualize it.

For code, authors state use of publicly available algorithms, but it would help assessing analytical steps taken if the scripts/code used for this particular data analysis were provided as a supplementary material.

We added a link in the methods section that refers to the code on the GITHUB of the laboratory which contains all the relevant code.

Figures:

Fig2B - it would be better to use color palette similar to 2C, to allow clear identification of cell populations from collected time-points. Gradual color palette used doesn't allow simple identification. This would also allow easier comparison with identified clusters in 2C.

Figure 1D has now a more defined palette

Figure 3J - Liu et al., 20207 should probably be just Liu et al., 2020? similarly in K), 20152 - should this be 2015? Probably reference number - should be upper index.

This has been fixed now.

Figure 4C - are there some labels overlaid over cells? With this size of the image, it is hard to distinguish, but there seem to be some artifacts visible in the middle of each cell. As these don't seem to serve here any purpose, it would be better to display original IF image without labels.

The reviewer is correct, the cells were numbered for counting and we agree that for a representative purpose they should be removed. The new Figure 6C is now free from labels.

Figure 4E - Anova with post-hoc Tukey test is indicated for this graph, although it compares only 2 conditions - was it really the test used?

The reviewer is correct, the test used to compare the two conditions reported in Figure 4E was the Mann-Whitney test. We corrected the caption accordingly (page 17).

Figure 4F,G - why was single color representation used for double immunostaining of NANOG/TRA-1-60? Only NANOG was displayed for cells reprogrammed in microfluidics device, it might be probably more useful to display only NANOG here as well.

In "open well" configuration, due to a less controllable environment and more asynchronous acquisition of pluripotency by each individual colony, it is likely to have a significant drop in the concordance between NANOG and TRA-160. For this reason we preferred to keep for this experiment both markers.

Figure 4G - should there be "temporally modulated"?

Well noted and added on Figure 6G (highlighted).

Reviewer #3

In this manuscript, the Authors exploit a microfluidic platform that they have spent years refining, and apply it to the definition of the paths chosen by cells towards human pluripotency. The Authors argue that this device allows them to define intercellular factors contributing to the generation of pluripotency, in line with what has been already shown in mouse cells.

On the overall, I think the approach and the findings are intriguing. The suggestion that apparently "dead end" populations may instead be conductive of the reprogramming is very interesting. I feel however that a few key aspects should be properly addressed, before recommending publication in Nature Communication. Specifically, I understand that the microfluidic approach is very useful in increasing the yield of the generated iPS, based also on the previous publications of the group, including the Nature Protocols papers. However, the Authors in this manuscript aim to uncover the biological underpinning of reprogramming. As such, several choices that are made, including the media and growth factors, the experimental settings and more broadly the small and big experimental choices that were made, question the physiological relevance of their findings. I think that a much more detailed methodological description should be included, and most importantly, appropriate positive and negative controls added, to assuage the concern that their findings could be artifactual, or at least driven by the

chosen experimental settings. Just to provide an example, in line 89 is mentioned that "A medium that contains only 3 types of proteins (FGF2, INS, TF) preserved the high efficiency of microfluidic reprogramming". How this choice would affect the reprogramming? if some of these factors were to be replaced, or else others added or removed, would be the specific and overall conclusions of the paper maintained?

We thank the reviewer for his/her interest in our work, and we address the comments as follows.

We added new figures and experimental details describing the microfluidic reprogramming process in material and methods (Page 9).

We understand the reviewer's comment that microfluidics reprogramming can contain some artifact phenomena. However, to avoid any doubt regarding this, first we explain the rationale of using microfluidic as discovery tools for detecting endogenously secreted proteins that enhance reprogramming and, second, we fully focus in the gain of function experiments using discovered ligands in standard conventional culture system (multi well), as reported in revised Fig.6 (old Fig.4).

The use of microfluidic system as valuable tool to accumulate endogenous secreted proteins in a confined environment was recently reported by us in Michielin et al, 2020 (PMID: 32826061); In general, we can speculate that factor accumulation in a confined environment seems more consistent with physiological conditions, where the ratio between interstitial cell volume and volume of the cells is quite low. This condition is more represented in microfluidics rather than standard wells, where the factors secreted by the cells are diluted in the media.

On the other hand, the gain of function is very informative because showed that the identified ligand accumulation increases the reprogramming efficiency in conventional culture system. We also showed that the microfluidics approach overall recapitulates analog reprogramming systems as all the transitions, signatures and hallmarks are perfectly in line with another independent human secondary system we previously characterize (Cacchiarelli et al, 2015 - PMID: 26186193).

Given that, we discussed the idea of using microfluidics to specifically probe a cross-talk between specific populations that is hidden in conventional systems. We rephrased some parts of the manuscripts to fully display the novelty without any misleading message (Discussion page 7). Regarding the sentence in line 89 we wrote as follow:

"To maximise the effectiveness of identifying the endogenous secreted proteins we used a chemically defined medium based on E6 medium, with the addition of FGF2 , which has been shown to preserve the high efficiency of microfluidic reprogramming (Extended Data Fig. 1A) while enabling high-resolution and accurate detection of cell-secreted proteins (Extended Data Fig. 1B-D and Methods)." (highlighted on page 2).

Minor points:

- a more detailed description of the microfluidic platform used should be included in the material and methods section. Also. figure 1A is not very informative. I would replace the fig1A with extended figure 1A, and with a more detailed description on the corresponding figure legend (now limited to "A) Schematic representation of the experimental design for the proteomic study."). Without reading the previous technical papers from the same authors, it is difficult to understand the methodology.

We added new paragraphs in Material and methods section, named microfluidic device and reprogramming in microfluidics, to describe the microfluidic platform used in this study and the procedure for performing reprogramming in microfluidics (Page 9).

Moreover, in Fig. 1A-B we added a schematic representation of both microfluidic and sampling method for a non-expert reader for both proteomic and scRNA-seq. Since the specifics of the two methodologies are different, we are keeping the details in the Methods sections.

- what is the meaning of "Protein tagging allowed us to obtain a relative quantification of each protein along the process (Extended Data Fig. 1A)."? In the methods and figure legends this information should be included.

We thank the reviewer for this suggestion. We integrated both the Results section (Page 2) and Figure 1B caption for better clarity. We also added the following reference on the methodology of relative quantification by TMT labeling (Thompson et al, 2003 - PMID: 12713048).

- It is unclear the statement "We quantified 4542 proteins, identified in either two (19%) or three replicates (81%)." I would imagine a higher number from 2 replicates.

This statement is correct, most of the proteins were identified in all 3 replicates, only fewer (19%) were missing in one of the replicates. We rephrased it for clarity into "We quantified 4542 proteins, the majority identified in 3 replicates (81%) and the others identified in only 2 replicates." (Highlighted on page 2).

- I would ask the Authors to show the data reported in "Therefore, as NRG1-ERBB3 expression occurs only along the reprogramming trajectory, we did not get significant results when comparing MEFs versus human reprogramming intermediates from our human secondary system (data not shown)", line 246.

We added this data in Extended Data Fig. 5E to show this comparison.

REVIEWERS' COMMENTS

Reviewer #1 (Remarks to the Author):

The authors addressed most of my concerns and this version of the manuscript was greatly improved. I can recommend it for publication.

Reviewer #2 (Remarks to the Author):

Panariello, Gagliano et al. used a microfluidics facilitated cellular reprogramming to analyze proteins secreted into the medium during this process by LC-MS/MS proteomic analysis to gain insight into the pathways participating in reprogramming and the role of different cell populations. In this revised manuscript, authors addressed most of the concerns raised by the reviewers, which in my view resulted in more complete and clearer paper. I recommend it for publication in Nature Communications.

One small new issue, on lines 696-698: New text explaining use of TRA1-60 and NANOG as markers seems to be incomplete:

"Since TRA1-60 is a membrane/extracellular marker and NANOG, we considered TRA1-60 area positive only where it overlapped with NANOG positive nuclear area, for having the double positive cells as result."

Should this be "and NANOG is a nuclear marker,..."?

Reviewer #3 (Remarks to the Author):

The Authors have addressed by points.

Response to reviewers

Reviewer #1 (Remarks to the Author):

The authors addressed most of my concerns and this version of the manuscript was greatly improved. I can recommend it for publication.

Reviewer #2 (Remarks to the Author):

Panariello, Gagliano et al. used a microfluidics facilitated cellular reprogramming to analyze proteins secreted into the medium during this process by LC-MS/MS proteomic analysis to gain insight into the pathways participating in reprogramming and the role of different cell populations. In this revised manuscript, authors addressed most of the concerns raised by the reviewers, which in my view resulted in more complete and clearer paper. I recommend it for publication in Nature Communications.

One small new issue, on lines 696-698: New text explaining use of TRA1-60 and NANOG as markers seems to be incomplete:

“Since TRA1-60 is a membrane/extracellular marker and NANOG, we considered TRA1-60 area positive only where it overlapped with NANOG positive nuclear area, for having the double positive cells as result.”

Should this be “and NANOG is a nuclear marker, ...”?

We thank the reviewer for noticing. We added this specification where suggested.

Reviewer #3 (Remarks to the Author):

The Authors have addressed by points.